# Pulse irradiation synthesis of metal chalcogenides on flexible substrates for enhanced photothermoelectric performance

Yuxuan Zhang [1], You Meng [2] ✉, Liqiang Wang[3], Changyong Lan[4], Quan Quan[1], Wei Wang[1], Zhengxun Lai[1], Weijun Wang[1], Yezhan Li[1], Di Yin[1], Dengji Li[1], Pengshan Xie[1], Dong Chen[1], Zhe Yang[5], SenPo Yip[6], Yang Lu [7], Chun-Yuen Wong [5] ✉ & Johnny C. Ho [1,2,6] ✉

High synthesis temperatures and specific growth substrates are typically required to obtain crystalline or oriented inorganic functional thin films, posing a significant challenge for their utilization in large-scale, low-cost (opto-)electronic applications on conventional flexible substrates. Here, we explore a pulse irradiation synthesis (PIS) to prepare thermoelectric metal chalcogenide (e.g., $Bi_2Se_3$, $SnSe_2$, and $Bi_2Te_3$) films on multiple polymeric substrates. The self-propagating combustion process enables PIS to achieve a synthesis temperature as low as 150 °C, with an ultrafast reaction completed within one second. Beyond the photothermoelectric (PTE) property, the thermal coupling between polymeric substrates and bismuth selenide films is also examined to enhance the PTE performance, resulting in a responsivity of 71.9 V/W and a response time of less than 50 ms at 1550 nm, surpassing most of its counterparts. This PIS platform offers a promising route for realizing flexible PTE or thermoelectric devices in an energy-, time-, and cost-efficient manner.

Scalable thin film deposition techniques, including chemical (physical) vapor deposition and solution-processed coating, have been developed to produce amorphous or nanocrystalline films[1–5]. However, additional high-temperature processing steps (typically above 400 °C) are necessary when crystalline films are required[6,7]. This requirement can pose significant challenges in achieving desirable features of crystalline films while working with thermally unstable substrates and other device components[8–10]. In particular, polymeric substrates, which are extensively used in flexible electronics, are heat-intolerant and atomically disordered, rendering them incompatible with high-temperature treatment and high-quality material growth.

Consequently, the performance of electronic devices fabricated on such substrates is frequently compromised by the presence of inferior crystallinity and lattice defects[4,11–13]. Thus, there is a pressing demand for thin-film processing methods that enable direct low-temperature deposition while still providing high-quality semiconductor characteristics.

Although flexible substrates are often deemed simply as platforms to support device components, their properties can significantly impact the overall device performance if their thermal impacts take effect. Unlike solid-state inorganic materials, polymeric matrixes usually consist of disordered long chains with low crystallinity, giving

[1]Department of Materials Science and Engineering, City University of Hong Kong, Hong Kong SAR 999077, P.R. China. [2]State Key Laboratory of Terahertz and Millimeter Waves, City University of Hong Kong, Hong Kong SAR 999077, P.R. China. [3]Department of Mechanical Engineering, City University of Hong Kong, Hong Kong SAR 999077, P.R. China. [4]State Key Laboratory of Electronic Thin Films and Integrated Devices, University of Electronic Science and Technology of China, Chengdu 610054, P.R. China. [5]Department of Chemistry, City University of Hong Kong, Hong Kong SAR 999077, P.R. China. [6]Institute for Materials Chemistry and Engineering, Kyushu University, Fukuoka 816 8580, Japan. [7]Department of Mechanical Engineering, The University of Hong Kong, Hong Kong SAR 999077, P.R. China. ✉e-mail: youmeng2@cityu.edu.hk; acywong@cityu.edu.hk; johnnyho@cityu.edu.hk

rise to a thermal insulating characteristic[14,15]. In this regard, the temperature distribution on polymeric substrates is susceptible to small amounts of energy input. Therefore, rational substrate thermal management, such as selecting a flexible substrate with suitable thermal properties, can be a promising strategy for designing thermal-mediated devices. For example, detectors based on the PTE effect, which arises from the photo-induced thermal effect and the Seebeck effect, are well known for their broadband responses[16]. Previous studies have reported that the responsivity of PTE detectors can be improved by utilizing electrical gating, surface plasmonics, antenna coupling, and phonon absorption, which result in a higher temperature gradient or enhanced Seebeck coefficient[17–21]. In the meantime, a prior work has pointed out that the underlying solid substrates can also influence the PTE response via thermal effects[22]. Thus, unveiling the thermal coupling effect between the unexplored polymeric substrates and functional films is essential to improve device performance.

Herein, we effectively lower the processing temperature of metal chalcogenides (e.g., $Bi_2Se_3$, $Bi_2Te_3$, and $SnSe_2$) down to 150 °C via a pulse irradiation process, unachievable in other conventional techniques. By carefully tuning the irradiation and temperature profiles, and the structure of the reactant layer, we achieve an ultra-short reaction time of less than one second, demonstrating the energy efficiency of PIS. It is proposed that the self-propagating combustion process is responsible for ultrafast low-temperature synthesis. Compared to non-PTE photodetectors, our PTE-based film photodetectors produced by PIS exhibit good compatibility with flexible substrates and bandgap-independent wide spectral response. Importantly, we discover a synergistic thermal effect between polymeric substrates and PTE films, which offers an alternative route for improving the PTE with wide-band detection capability.

## Results

### $Bi_2Se_3$ thin-film characterization

The $Bi_2Se_3$ film is prepared by stacked elemental layers with a subsequent pulse irradiation synthesis (Fig. 1, see Methods for detailed information). Specifically, selenium (Se) and bismuth (Bi) layers are sequentially deposited in an ultra-high vacuum physical vapor deposition chamber. The Bi layer is always deposited atop the Se layer to prevent Se loss during PIS[23]. In a typical PIS process, the high-intensity radiation generated from the tungsten halogen lamp heats the sample, and the temperature rises to a preset point within a few seconds. The precisely controlled temperature system enables fast ramping rates (-100 K s⁻¹) with a heating duration of less than one second (upper panel of Fig. 1b). Consequently, the pulse irradiation triggers the interdiffusion between layers, and $Bi_2Se_3$ nucleates and grows into a layered structure. In the meantime, a mass flow of inert Ar is used to prevent the film from possible oxidation.

Grazing-incident wide-angle X-ray scattering (GIWAXS) patterns in Fig. 2a–d reveal that the film has a uniaxial texture with a crystallographic c-axis aligned perpendicularly to the substrate surface. Even if the PIS temperature is as low as 150 °C, $Bi_2Se_3$ still emerges with diffractions of (003) and (006) at $q_z = 6.6$ nm⁻¹ and 13.2 nm⁻¹ (Fig. 2a)[24]. The preferential orientation becomes prominent as the PIS temperature increases to 300 °C (Fig. 2b–d and Supplementary Fig. 1), which suggests that the prime crystallite extends along the substrate surface and the grain sizes increase with temperature (Supplementary Fig. 2). The structural evolution is detailed in Raman spectra (Supplementary Fig. 3 and Supplementary Note 1) and X-ray photoelectron spectroscopy (Supplementary Fig. 4)[25]. High-resolution transmission electron microscopic (HR-TEM) images further reveal the transformation process. Figure 2e illustrates clear interfaces between Bi and Se elemental layers before PIS. Corresponding selected-area electron diffraction patterns (SAED) show the halo feature with diffraction rings, which are associated with amorphous Se phases and tiny polycrystalline Bi layers (Fig. 2f). After PIS, the well-crystallized $Bi_2Se_3$ film is obtained, and the layered structure is as well distinguishable in the SAED image (Fig. 2g, h). It is noted that grain boundaries can be observed from a larger view, suggesting that the film is not a perfect single crystal (Supplementary Fig. 5).

Further insight into the layered structure crystallization during PIS can be obtained through the density functional theory-based molecular dynamic (MD) simulation. In a simulated annealing process of the Bi/Se layered structure, typical snapshots during MD trajectories show the interdiffusion that initiates from the interface and a rapid fusion that almost finishes after 18 ps (Fig. 2i and Supplementary

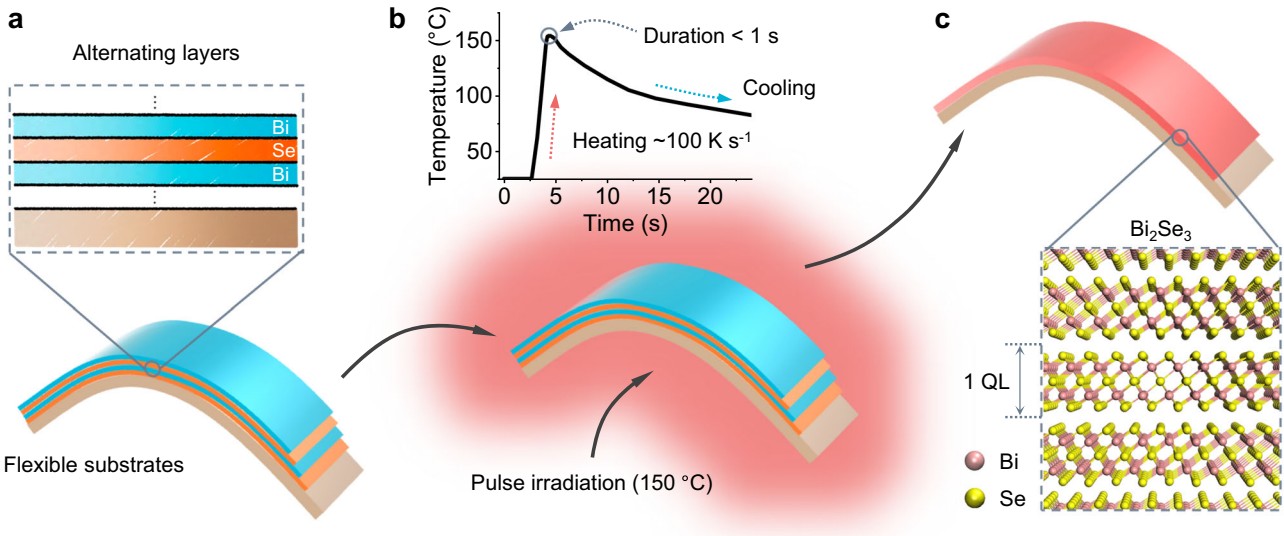

**Fig. 1 | Schematic of pulse irradiation synthesis (PIS) for highly conformable photothermoelectric (PTE) films. a** In PIS, alternating elemental layers are initially deposited on flexible substrates. **b** The film is then subjected to pulsed irradiation, which induces rapid heating of the film at a rate of approximately 100 K s⁻¹. The temperature profile in a typical PIS is shown in the upper panel. **c** Owing to the rapid heating, crystallization is initiated at a significantly lower temperature (e.g., 150 °C), thereby preserving the flexibility of the substrates. Quintuple layers (QL) of the $Bi_2Se_3$ crystal structure are shown in the lower panel.

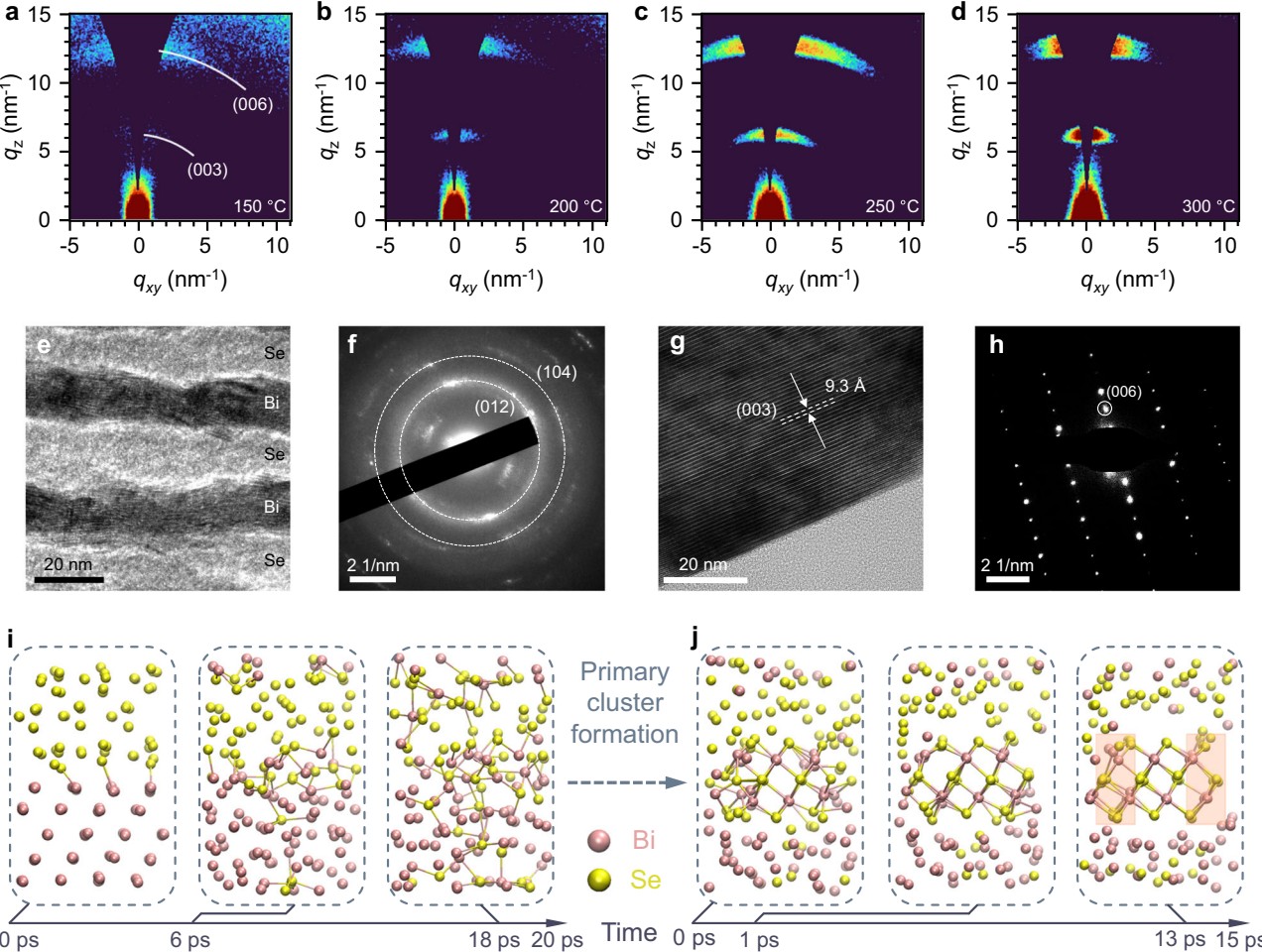

**Fig. 2 | PIS synthesis of large-area PTE film and characterization. a–d** Grazing-incident wide-angle X-ray scattering (GIWAXS) patterns of $Bi_2Se_3$ films after PIS at 150 (**a**), 200 (**b**), 250 (**c**), and 300 °C (**d**) for 1 s, respectively. White arcs in (**a**) represent diffractions of $Bi_2Se_3$ with a preferential out-of-plane orientation. **e, f** High-resolution transmission electron microscopic (HRTEM) image and selected-area electron diffraction (SAED) pattern of the film after evaporation. The white dashed rings in (**f**) correspond to diffractions of polycrystalline Bi layers.

**g, h** HRTEM image, and SAED pattern of the film after 300 °C PIS, showing the (003) interplanar spacing (**g**) and (006) diffraction spot of $Bi_2Se_3$ (**h**). **i, j** Typical snapshots taken in the ab-initio molecular dynamic (AIMD) simulation during the interdiffusion of Se and Bi layers (**i**) and attachments of Se and Bi atoms to the edge of the primary $Bi_2Se_3$ cluster (**j**), almost completing two columns of atoms on both sides (highlighted with colored boxes).

Movie 1). Once the primary cluster is formed, Bi or Se atoms approach the edge of the $Bi_2Se_3$ cluster and promote its lateral crystal growth (Supplementary Movie 2). These atoms almost complete two new rows of $Bi_2Se_3$ after 1 ps, while the layered structure is further stabilized after 13 ps (Fig. 2j). This supports the energy-favorable PIS-mediated rapid nucleation and lateral growth of layered $Bi_2Se_3$.

**Mechanism of pulse irradiation synthesis**

A self-propagating combustion process is proposed for this ultrafast and low-temperature synthesis. In this work, the exothermic reaction of $2Bi + 3Se \rightarrow Bi_2Se_3$ exhibits a negative formation enthalpy (~ −140 kJ mol⁻¹), and the calculated adiabatic temperature is ~722 °C, which is much higher than the melting temperatures of both Bi (271.4 °C) and Se (220.8 °C), satisfying the prerequisite for sustaining the propagation of the combustion wave[26]. Upon ignition of the stacked Bi/Se layers by an energy pulse, the heat released from the exothermic reaction outpaces the heat dissipation rate from the reactants, causing the local temperature to rise near the adiabatic temperature. Thus, a combustion wave passes through the film as the self-generated heat is sufficient to maintain the reaction until the complete consumption of the reactants. This phenomenon eliminates the need for extra energy, significantly reducing the processing temperature. Furthermore, the

ignition temperature is confirmed by the differential scanning calorimetry (DSC) curve, where the exothermic peak indicates that the ignition temperature is 150 °C (Supplementary Fig. 6a), consistent with previous observations. Interestingly, the ignition temperature in this work is lower than that of conventional combustion synthesis of bulk metal chalcogenides (generally 220–350 °C)[27–29], which can be attributed to the kinetic process (i.e., thermal diffusion)[30]. We estimate the characteristic diffusion distance of Se using the one-dimensional semi-infinite model, yielding a result of ~1.1 μm (Supplementary Note 2), which covers the entire thickness of the ultrathin elemental film (10 nm). Combined with a small formation barrier of 0.11 eV (Supplementary Fig. 6d), the self-propagating combustion can be triggered at a relatively low temperature and completed within a few seconds via PIS.

With the low-temperature PIS, the rapid heating/cooling process helps bypass the thermal-induced degradation of flexible substrates. After the conventional furnace annealing (CFA) at 150 °C, the polyethylene terephthalate (PET) deforms and losses its integrity, while no deformation is observed after PIS at the same temperature (Supplementary Fig. 7). Furthermore, the PIS yields a dense, flat, and continuous film as confirmed by scanning electron microscopy (SEM), which is in contrast to the CFA treated film with apparent porosity and

cracks (Supplementary Fig. 8a–e). At the same time, the PIS also avoids the Se sublimation loss due to its high saturated vapor pressure, which is confirmed by the compositional depth profiling of the film (Supplementary Fig. 8f, g)[31]. For the same reason, the film thickness remains nearly unchanged after PIS (Supplementary Fig. 9). Surface morphologies are then examined by atomic force microscopy (AFM), revealing that PIS-treated samples exhibit a smooth surface, with the standard deviation of roughness ($R_a$) slightly increasing from 2.60 nm (150 °C) to 3.36 nm (400 °C) as the temperature rises (Supplementary Figs. 10 and 11).

### Thermal coupling between substrates and PTE films

After that, the thermal coupling effect between polymeric substrates and metal chalcogenide films is explored. We find that the photoresponsivity is closely related to the magnitude of the temperature gradient within the substrate, and the thermal diffusivity ($\alpha$) of the

substrate is the main factor that influences the temperature ($T$) distribution (Supplementary Note 3), which can be described by:

$$\frac{\partial T}{\partial t} = \alpha \nabla^2 T \tag{1}$$

Figure 3a, b illustrates the simulated temperature distribution with a device configuration shown in Fig. 3c. Due to the significantly different thermal diffusivity (0.21 and 0.87 mm² s⁻¹ for PI and SiO₂, respectively), a temperature distribution of −0.317 K μm⁻¹ is found on PI, larger than −0.036 K μm⁻¹ of SiO₂ (Fig. 3d, e). Accordingly, the different temperature gradients lead to a larger photoresponse of the PI-based device (1.20 mV) than the SiO₂-based one (0.08 mV) (Fig. 3f). Benefiting from facile deposition, the PIS is readily compatible with multiple polymeric substrates, such as polydimethylsiloxane (PDMS), PET, and poly(ethylene 2,6-naphthalate) (PEN). The compiled

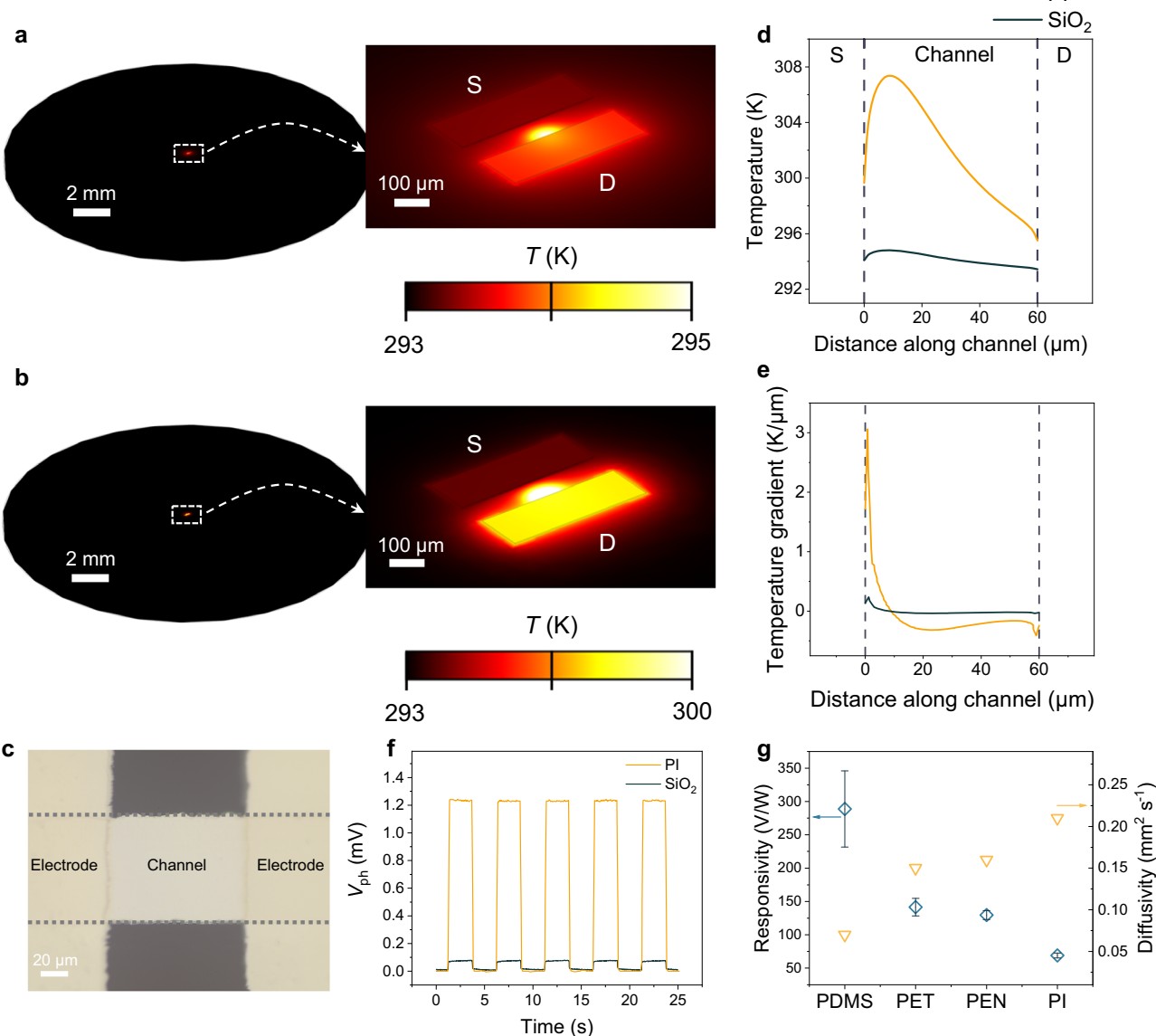

**Fig. 3 | Thermal management of polymeric substrates. a, b** Thermal simulations on SiO₂ (**a**) and PI (**b**) surfaces based on the device architecture. S and D denote source and drain electrodes, respectively. **c** Optical image of the Bi₂Se₃ film photodetector. **d, e** Corresponding temperature (**d**) and gradient (**e**) profiles across the device channel. Two vertical dashed lines indicate the interfaces between electrodes and channels. **f** Temporal photovoltage ($V_{ph}$) response curves comparing the

PTE film on Si/SiO₂ and PI substrates. **g** Responsivity on four typical flexible substrates, i.e., polydimethylsiloxane (PDMS), polyethylene terephthalate (PET), poly(ethylene 2,6-naphthalate) (PEN), and polyimide (PI), with the summary of the corresponding diffusivity (Supplementary Table 1), indicating an inverse correlation between them. Error bars indicate standard deviations of responsivities obtained from 30 individual devices.

performance is also consistent with the simulation result (Supplementary Fig. 12). Therefore, using substrates with lower thermal diffusivity can effectively improve the photovoltages and responsivities (Fig. 3g and Supplementary Table 1). It is noted that all flexible substrates show poor absorption with minor differences in absorption spectra, whereas the $Bi_2Se_3$ film demonstrates higher absorption (Supplementary Fig. 13), which further underscores that the device performance is primarily governed by the inherent thermal properties of the underlying substrate itself, instead of its optical absorption. In addition, surface meta-materials can be employed as plasmonic absorbers to improve performance[32]. It is observed that the photoresponse increases ~60% after integration with gold-based metamaterials (Supplementary Fig. 14). This result may prompt further investigations into the synergistic effect of thermal coupling and surface plasmonic absorption.

## PTE response of metal chalcogenide films

For PTE devices, when the laser focuses on one side of the channel, the temperature gradient is built, which drives the diffusion of electrons from the warm side to the cold side (major carriers in $Bi_2Se_3$ are electrons). As a result, we can observe a negative electrical potential (Fig. 4a). Unlike traditional photoconduction detectors, the PTE photovoltage can be changed or even reversed by altering the temperature gradient along the channel through manipulation of laser illumination position. The scanning photovoltage mapping (SPVM) directly exhibits an opposite photovoltage ($V_{ph}$) of ±180 μV at two ends of the channel (Fig. 4b). Meanwhile, a zero $V_{ph}$ crossover position is found located at the center area of the channel. Such zero-$V_{ph}$ zone can be shifted when an external bias ($V_{ds}$) is applied and finally disappear at a $V_{ds}$ of 200 μV (Fig. 4c and Supplementary Fig. 15), agreeing well with the typical PTE

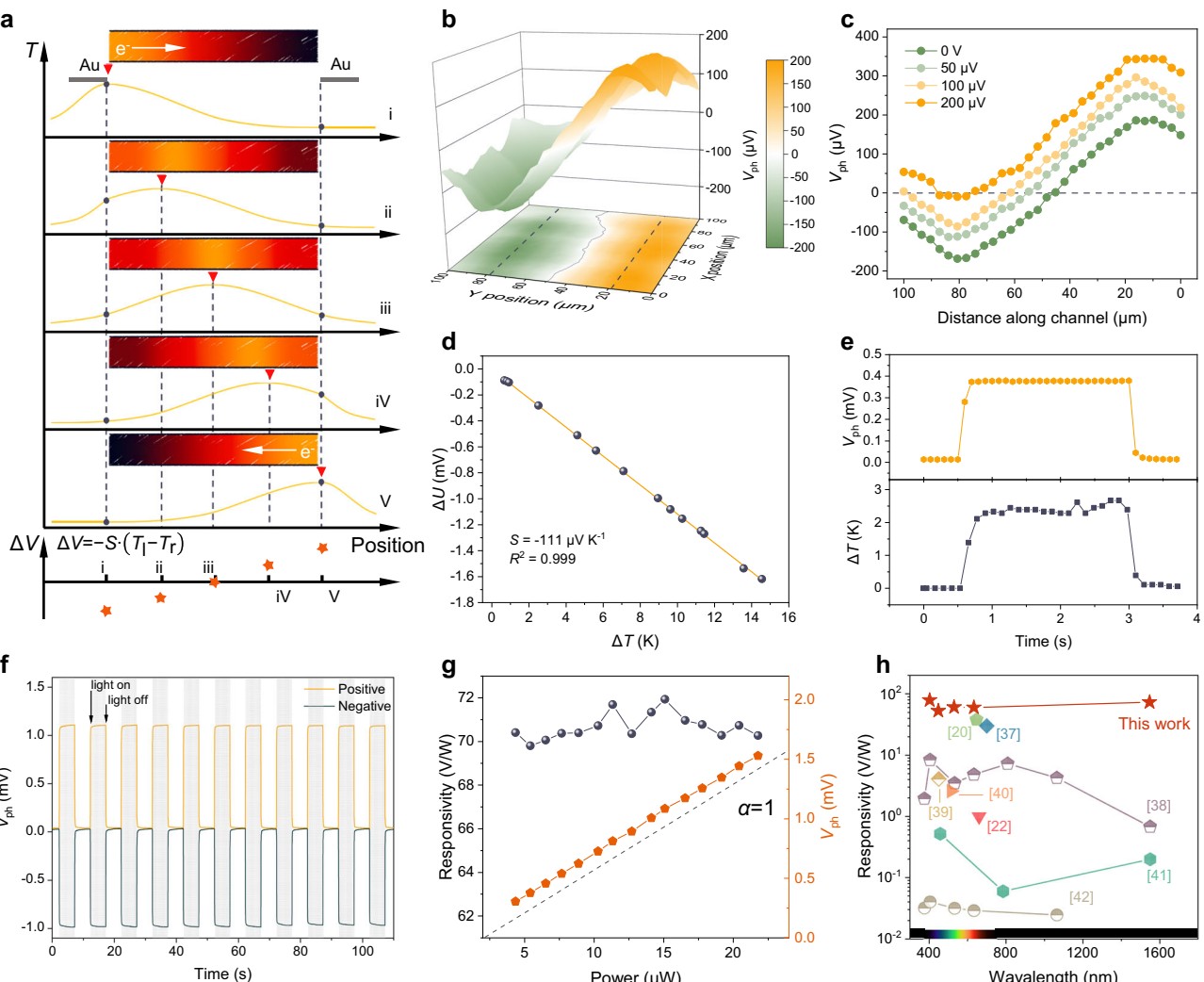

**Fig. 4 | Mechanism and performance of PTE photodetectors. a** Schematic of the local laser-induced temperature distribution $T$ and potential distribution $\Delta V$ along the channel when the laser spot moves from the left electrode to the right one. Red triangle cursors indicate the position where the focused laser is illuminated in the device channel. The white arrow indicates the diffusion direction of hot electrons. $S$, $T_l$, and $T_r$ refer to the Seebeck coefficient and temperatures on the left and right electrodes, respectively. **b** Scanning photovoltage mapping (SPVM) in a 3D plot. Values projected on the bottom surface show a symmetric distribution of $V_{ph}$. Gray dashed lines indicate the channel-electrode interface and the solid grey line refers to the contour with zero $V_{ph}$. **c** Line profiles of the $V_{ph}$ cutting along the channel under increasing $V_{ds}$ with the dashed line indicating the zero $V_{ph}$. **d** Voltage across the $Bi_2Se_3$ film versus the corresponding temperature difference to determine its room-temperature Seebeck coefficient $S$. The orange line is the linear fitting of the experimental data, with a coefficient of determination being 0.999. **e** Comparison of the $V_{ph}$-time curve and the $\Delta T$-time curve. **f** Temporal response with the laser spot positioned on either side of the device, showing a positive and a negative photoresponse. **g** Responsivity and $V_{ph}$ as a function of incident power under 1550 nm illumination. The dashed line is a reference line whose ideal factor $\alpha$ equals to 1. **h** Benchmark of the photodetector of reported PTE detectors.

mechanism. According to the Seebeck effect, the photovoltage is generated by photo-induced temperature difference ($\Delta T$) as:

$$V_{ph} = -S \cdot \Delta T \tag{2}$$

where $S$ is the Seebeck coefficient. It characterizes the ability of a material to convert a temperature difference ($\Delta T$) to an electric potential difference ($\Delta U$)[33], and it is extracted to be -111 μV K$^{-1}$ (Fig. 4d), comparable with other $Bi_2Se_3$ films synthesized with conventional methods (Supplementary Table 2) and other chalcogenide counterparts[34–36], but with a lower processing temperature and a shorter processing time. It is worth mentioning that the unique photoresponse behavior of PTE distinguishes it from other mechanisms, such as the photovoltaic effect and photo-Dember effect (Supplementary Note 4). Recording by a thermal infrared camera (Supplementary Movie 3), a 1550 nm laser irradiation can induce ~3 K temperature difference on the channel surface, by which a $V_{ph}$ of ~0.4 mV is produced simultaneously (Fig. 4e), corresponding well with the Seebeck coefficient test (Fig. 4d). In addition, the PTE performance correlates with the fabrication process. The appropriate film thickness and processing temperature lead to an improved crystallinity and an optimized output $V_{ph}$ (Supplementary Fig. 16 and Supplementary Note 5).

Next, we systematically investigate PTE detectors on the PI substrate (Supplementary Fig. 17). The temporal $V_{ph}$ exhibits bi-directional positive/negative responses with well-symmetric characteristics when the 1550 nm laser illuminates two ends of the channel (Fig. 4f and Supplementary Fig. 18). The rapid and reproducible response under cycling switches indicate good stability in the near-infrared detection, with a high on/off ratio of ~10$^2$ (Supplementary Fig. 19). As we raise the light power ($P$), the $V_{ph}$ is proportionally increased (Fig. 4g and Supplementary Fig. 20), while the extracted channel resistances keep constant (Supplementary Fig. 21). The linear relationship between $P$ and $V_{ph}$ can be fitted by the power law equation: $V_{ph} \propto P^\alpha$, where the fitted ideal factor $\alpha$ is close to 1. These results rule out the possibility of photoconductive and bolometric effects contributing to photoresponse in this work. The corresponding responsivity ($R = V_{ph}/P$) remains at the same level with different $P$ and peaks at 71.9 V/W for 1550 nm IR light, surpassing most PTE detectors (Fig. 4h and Supplementary Table 3)[20,22,37–42]. Given the zero bias in the measurement, the shot noise in the dark current is negligible, whereas the flicker noise and Johnson noise will dominate the dark current in the PTE device[43]. The power-dependent detectivity ($D^*$) extracted from the noise equivalent power (NEP) reaches a maximum value of $1.85 \times 10^7$ Jones at 1550 nm (Supplementary Figs. 22 and 23), comparable with some recently reported PTE photodetectors[44,45]. The performance of the $Bi_2Se_3$-based PTE detector can be further improved through material doping, rational electrode design, plasmonic nanoantenna, hybrid structure, and so on[46–50].

As an important figure-of-merit for photodetectors, the rise and decay times are extracted to be 49.7 and 49.5 ms, respectively, which indicates an estimated 3 dB cutoff at ~20 Hz (Supplementary Fig. 24a, b). The scale of the PTE response time reflects the phonon-dominated transport in the $Bi_2Se_3$ film and the thermal equilibrium rate that largely depends on the thermal conductivity and the heat capacity of materials[43]. Under external bias, the photoconductive effect starts to contribute to the response (Supplementary Fig. 24c, d), and carrier scattering and electronic trapping are responsible for the relatively longer response time. Besides, in Supplementary Fig. 25, after 5000 s of repeated illumination, continuous illumination for 120 s, or 6-month ambient storage, there is hardly any discernable degradation of the photovoltage, signifying its robust detection stability. Also, the device maintains a reproducible photoresponse characteristic, with no discernable degradation of responsivity (~5%) after 5000 bending cycles (Supplementary Fig. 26).

Based on the mechanism of PTE, the detection wavelength in our proposed strategy is not limited by the bandgap of the active film. The PIS-treated films can realize wide-spectrum detections, by which the average responsivities under 405, 450, 532, and 635 nm illuminations are 75.8, 55.1, 52.4, and 40.0 V W$^{-1}$, respectively (Supplementary Figs. 27 and 28). Finally, to show the broad generality of PIS, $SnSe_2$ and $Bi_2Te_3$, which are typical thermoelectric materials usually obtained under high temperatures[34,51], are fabricated on PI via 150 °C PIS. These films with PIS exhibit good PTE response under IR illumination (Supplementary Fig. 29).

## Discussion

In summary, a low-temperature pulse irradiation method is explored to produce metal chalcogenide films for flexible PTE devices. In a typical PIS, the self-propagating combustion process triggered by an energy pulse effectively avoids high temperatures, and the resultant combustion wave rapidly passes through the thin film, allowing the material to crystallize within one second. Powered by the self-propagating combustion process, the PIS achieves a low synthesis temperature of 150 °C and completes the ultrafast reaction within one second, leaving minuscule energy overhead. Also, the synergetic effect between polymeric substrates and bismuth chalcogenides films improves the PTE performance through substrate-related thermal management. It is found that thermal diffusivities of different substrates correlate with their temperature profile under local illumination, which couples with the thermoelectric film and affects the output photovoltages. The low temperature and rapid process in PIS allow us to use flexible substrates with suitable thermal properties. In this regard, we present a pathway that further manipulates the performance of PTE devices. Moreover, the wide spectrum photoresponse, high responsivity (71.9 V/W for 1550 nm), and fast response (<50 ms) surpass most of the counterparts. The PTE effect, which includes two processes: the photothermal conversion and the Seebeck effect, provides a powerful platform that transfers optical signals to electrical readouts without the limitation of the bandgap of active materials. Thanks to the low processing temperature, PIS can be well generalized to a broad range of flexible substrates and materials, demonstrating its potential for the future development of flexible PTE photodetectors and thermoelectric devices.

## Methods

### PIS treatment and device fabrication

Elemental layers were thermally evaporated on the substrate in an alternative manner. For the $Bi_2Se_3$ film, the evaporation rate of Bi and Se was controlled to be around 0.5 Å s$^{-1}$. The thicknesses of the Bi and Se layers were 10 and 15 nm, respectively. The pattern of the photodetection film was defined by a shadow mask. Then, the PIS was realized in a rapid thermal process system (AccuThermo AW410). A susceptor coated with inert silicon carbide was used to encase the substrate with evaporated films. The ramping temperature was ~100 K s$^{-1}$ with a steady time of less than 1 s. Thus, the synthesis time was 2 s and 4 s for PIS temperatures of 150 and 300 °C, respectively. To avoid the overshoot and undershoot of the temperature profile and ensure repeatability, all parameters were carefully tuned. During the treatment, high-purity Ar with a flow rate of 10 standard liters per minute (SLPM) was introduced into the system to protect the sample. Photodetectors were fabricated using metal lithography stencils to deposit the source and drain electrodes. The laser-cut mask with a distinct channel length of 60 μm was manually aligned on top of the pre-patterned photodetector film via an optical microscope. A 50 nm gold electrode layer was then deposited through thermal evaporation (rate: 1 Å s$^{-1}$).

### Material characterization

The texture of the film was characterized by GIWAXS using a small-angle X-ray scattering system (Xeuss 3.0). The sample-detector

distance was set to 120 mm, and the exposure time was 1800 s. The phase composition was studied from the XRD (D2 Phaser X-ray Diffractometer System). The surface morphology was characterized by SEM (Quanta 450), and the crystallography structure was determined by TEM (CM-20, Philips). The film topography and the roughness were evaluated by AFM (MultiMode 8, Bruker). The Raman spectra were measured by the confocal microscope spectrometer (Alpha 300R, WITec). For the investigation of the surface chemical state and the atomic ratio of the film, XPS was employed using a Thermo Fisher ESCLAB Xi$^+$ system. The depth profiling of the elemental ratio was realized by Ar$^+$-etching, with each etching depth controlled to approximately 10 nm. The thermal analysis during heating was realized by a differential scanning calorimeter (DSC 8000, PerkinElmer). A stoichiometric amount of high-purity Bi and Se powder was weighted and mixed in an agate mortar and then transferred into an aluminum pan, which was then cold pressed and sealed via a universal crimper press. In order to approximate the PIS process as closely as possible, the pellet was heated at a rate of 100 K min$^{-1}$. The Seebeck coefficient was obtained from a thermoelectric material test system (CTA-3, Cryoall). Before measurement, two thermocouples were connected to the film to obtain the voltage-current curves at room temperature. The test proceeded when the coefficient of determination for the fitted voltage-current curve exceeded 0.99999, which indicated good contact of the thermocouple with the surface. A heater was employed to create a temperature gradient in the film. Once the temperature difference stabilized, temperatures at the two thermocouple locations and the electrical potential between them were measured simultaneously. Then, the Seebeck coefficient was extracted from the $\Delta U$-$\Delta T$ curve (Fig. 4d).

## Device measurement

The temporal photovoltage and electrical performance of the device were recorded by the Agilent 4155 C semiconductor analyzer with a standard electrical probe station. The lasers (405, 450, 532, 635, and 1550 nm) were guided by an optical fiber and focused on the device with a microscope system, and were calibrated by a power meter (PM400, Thorlabs). For 405, 450, 532, and 635 nm lasers, a homemade light chopper and an attenuator were used to control the light and its intensity, respectively. While for 1550 nm laser, a modulator (AFG 2005, Arbitrary Function Generator, Good Will Instrument Co. Ltd) attached to the laser generator was used to modulate the IR irradiation.

The detectivity ($D^*$) was extracted as:

$$D^* = \frac{\sqrt{A\Delta f}}{\text{NEP}} \tag{3}$$

where $A$ is the device area, and $\Delta f$ is the bandwidth. The NEP was calculated by NEP = $v_n/R$, where $v_n$ is the measured voltage noise (Supplementary Fig. 22), and $R$ is the responsivity. To determine the response time, the high-resolution photovoltage curve was recorded by a highly sensitive source/measurement system (B2912A, Keysight). Rise and decay times are defined as the time interval of the net voltage change between 10% and 90%, respectively. The temperature change on the film was recorded by an infrared camera (Ti480 PRO, Fluke). Specifically, this test was carried out with a larger beam size (~1.5 mm) to ensure it exceeded the minimum resolution of the infrared camera. Moreover, the measurement was synchronized with the change of photovoltage with and without illumination.

## SPVM measurement

To investigate the working mechanism of the PTE detector, we characterized the photovoltage mapping by an integrated optoelectronic scanning system (ScanPro Advance, Metatest). The spot size is controlled to be 3 μm to ensure a high spatial resolution of the mapping.

Before scanning, a light power meter was placed directly below the sample to test the actual light power applied to the device (3.4 μW). The photovoltage was monitored by a source/measure unit (2636B, Keithley).

## Density functional theory (DFT) calculation

Ab initio calculation was carried out within the DFT framework with Vienna Ab-initio Simulation Package (VASP)[52,53]. The core level electron wave functions were expanded with the full-potential projected augmented wave (PAW) method[54] and the generalized gradient approximation (GGA) of Perdew-Burke-Ernzerhof (PBE) exchange-correlation functional was used to describe the interaction between electrons[55]. The cutoff energy for the plane wave expansion was set to 400 eV. The convergence criteria for the force of the structure and the energy of the self-consistent calculation were set at $10^{-2}$ eV Å$^{-1}$ and $10^{-5}$ eV, respectively. A vacuum layer of 30 Å was built for all structures to avoid perturbation between neighboring layers. The transitional state was searched and the energy barrier was calculated using the CI-NEB[56] with a force threshold of 0.03 eV Å$^{-1}$. The Brillion zone was sampled by a $3 \times 3 \times 1$ grid mesh. The molecular dynamic simulation was achieved by the ab-initio molecular dynamic (AIMD) calculation with VASP. Considering the calculation efficiency, the energy cutoff was reduced to 350 eV and the Brillouin zone was only sampled by Γ point. AIMD trajectories were run with the canonical ensemble (NVT) at the temperature of 573 K. The time step of the AIMD was 2 fs with an overall time span of 20 and 15 ps for interdiffusion and crystal growth, respectively. The central part of Se$_1$-Bi$_1$-Se$_2$-Bi$_1'$-Se$_1'$ was fixed to simulate the primary cluster during the simulation.

## Modeling and COMSOL simulation

Two substrates were constructed with PI and SiO$_2$. The silicon wafer used in this work was covered with 285 nm SiO$_2$. Thus, the bottom Si was omitted from the model. The finite element method (FEM) was carried out via COMSOL Multiphysics. The electrode was built on the substrate based on the actual size of the device. In the meantime, the PTE film on the substrates was only tens of nanometers, which indicated that the thermal distribution of the substrate dominates that of the film. Hence, we neglected the effect of the film on the heat transfer in the COMSOL simulation. For the boundary condition, a heat transfer coefficient of convection of 5 W m$^{-2}$ K$^{-1}$ was used and applied to all top surfaces, and the bottom surface of the simulation domain was fixed at room temperature (293 K). The density (SiO$_2$: 2200 kg m$^{-3}$; PI: 1300 kg m$^{-3}$), heat capacity (SiO$_2$: 730 J kg$^{-1}$ K$^{-1}$; PI: 1100 J kg$^{-1}$ K$^{-1}$), and thermal conductivity (SiO$_2$: 1.4 W m$^{-1}$ K$^{-1}$; PI: 0.15 W m$^{-1}$ K$^{-1}$) were extracted from the material database in the software. Furthermore, the surface emissivity was considered to calculate the temperature distribution (0.83 and 0.75 for PI and SiO$_2$, respectively). The deposited beam power density $P$ was quantified by Gaussian distribution as:

$$P = \frac{1}{2\pi r^2} \exp\left(-\frac{d^2}{2r^2}\right) \tag{4}$$

where $r$ is the radius of the laser, and $d$ is the distance between the light spot center and the specified point. The laser radius was 20 μm and the total power of the deposited light beam was 0.5 mW in the simulation.

## Data availability

Relevant data that support the key findings of this study are available within the article and the Supplementary Information file. All raw data generated during the current study are available from the corresponding author upon request. Source data are provided with this paper.

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

## Acknowledgements
This research was financially supported by a fellowship award from the Research Grants Council of the Hong Kong Special Administrative Region, China (CityU RFS2021–1S04) and the State Key Laboratory of Terahertz and Millimeter Waves (City University of Hong Kong).

## Author contributions
Y.Z., Y.M., and J.C.H. conceived the study and designed the experiments. Y.Z. and Y.M. carried out the sample preparation, XRD characterization, and device fabrication. L.W. and Y.Lu performed the TEM characterization. C.L. contributed to the SPVM measurement. Q.Q. contributed to the temperature measurement via the infrared camera. W.W. contributed to the XPS test. Z.L. performed the AFM and SEM measurements. Y.Li performed the DSC measurement. W.W., W.J.W., D.L., P.X. helped with the electrical measurement. Y.Z. performed the DFT calculation and COMSOL simulation. D.Y., D.C., Z.Y., S.P.Y., and J.C.H. contributed to the data analysis. Y.Z., Y.M., Y.Lu, C.Y.W., and J.C.H wrote and revised the manuscript.

## Competing interests
The authors declare no competing interests.
