## [Peer Review File · Nature Communications]

Pulse irradiation synthesis of metal chalcogenides on flexible substrates for enhanced photothermoelectric performanceREVIEWER COMMENTS

Reviewer #1 (Remarks to the Author):

In this manuscript, the authors prepared metal chalcogenide films on multiple polymeric substrates using pulse irradiation method. Furthermore, the photothermoelectric (PTE) performance can be enhanced due to the thermal coupling between thin films and polymeric substrates. There are several critical issues to be addressed before further consideration.

(1) The authors utilize the thermal coupling between the thermoelectric film and underlying substrate to tune the PTE detection performance. Similar strategy can be found in ACS Nano 2013, 7(8), 7271-7277. So, a proper discussion about the link between these two works should be included in the Introduction part.

(2) For the substrate-supported PTE detector, in addition to the substrate's thermal conductivity, its light absorption ability can also influence the photodetection performance remarkably. To clearly illustrate the contribution related with the substrate, the absorption spectra in the visible-near infrared regions of metal chalcogenide films and polymeric substrates (PDMS, PET, PEN, and PI) should be provided.

(3) In Figure 4b, the authors provided a scanning photovoltage mapping of Bi₂Se₃ PTE detector (Incident wavelength: 532 nm; Power: 0.5 mW; Spot diameter: 3 μm). The maximum photovoltage response is about 50 μV. As a result, the calculated responsivity is just 0.1 V/W, which obviously deviate from the claimed responsivity in Figure 4h (52.4 V/W at 532 nm). It is necessary to check the data consistency and accuracy.

(4) In Line 8 of Page 4, the authors claimed that the employment of PIS technique can avoid the Se sublimation. However, the EDX data in Supplementary Fig. 8f show that the element composition after PIS and CFA treatment is Bi₂Se_{0.6} and Bi₂Se_{2.5}, respectively. Please explain this inconsistency.

(5) The formula of specific detectivity (D*) is not correct (Line 6, Page 7). Usually, the D* is calculated using the following formula:

$$D^* = \sqrt{A} / NEP$$

$$NEP = \sqrt{4k_B T R} / RV$$

where k_B is the Boltzmann constant, T is the absolute temperature, and R is the device resistance.

(6) In Line 33 of Page 5 and Supplementary Figure 21, the authors mentioned the "photon thermal transport". What's the meaning of photon thermal transport?

(7) The authors attributed the fast response (around 50 ms) to the hot-carrier diffusion. However, this time is quite typical for thermal detectors where the phonon dominates the heat transport. For example, the response time of commercial infrared thermopile is usually around 20 ms. On the other side, the generated hot carriers approach thermal equilibrium with phonons at picosecond time scale, which also cannot support the author's conclusion.

(8) As a flexible photodetector, the bending or stretching stability is one of the important appraisal aspects. It would be better to provide the corresponding results.

(9) In the Abstract part, the wavelength for the 71.9 V/W responsivity should be provided.

(10) Some small mistakes should be corrected:

1) Line 8, Page 2, "...processing methods that enables direct low-temperature deposition...".

2) Line 27, Page 3, "The calculated adiabatic temperature is ~ 722 °C, which is much lower than the

melting temperature of both Bi (271.4 ° C) and Se (220.8 °C)".

Reviewer #2 (Remarks to the Author):

The authors have demonstrated a pulse radiation method to synthesize thermoelectric thin films. The low growth temperature requirement is quite suitable for flexible optoelectronic devices, directly growing on polymer substrates without degradation. Moreover, the low thermal diffusivities of these polymer substrates contribute to much enhanced PTE performance. This work is interesting and comprehensive, which should draw the attention of the researchers working in 2D materials synthesis, photodetectors, and thermoelectric materials. However, I have several questions about the measurement and find several problems in the manuscript, therefore this work should be reconsidered after answering the following questions:

1. The authors use the thermal camera to record the temperature difference under laser radiation. However, I think this measurement will not be accurate for small dimension devices, considering that the channel length is around 60 μm and the focus beam size is smaller than 3 μm . The temperature reading from the software should be the average of several pixel signals, and I doubt that the pixel resolution of a thermal camera can reach so high. I think the real temperature change induced by the laser should be even higher.
2. The Seebeck coefficient is derived from Fig. 4b. The author didn't provide information about how they got this figure. If the temperature change is also read from the thermal camera, then I will also question its accuracy. It is important to compare the Seebeck coefficients for PIS Bi₂Se₃ thin film with that synthesized with conventional methods. To get an absolute Seebeck coefficient, the authors can follow the methods in Nat. Commun. 2023, 14, 1938; Nano Letter. 2014, 14, 5, 2730.
3. The author claims to get a record PTE responsivity (71.9 V/W). However, the recent published Te nanowire on perfect absorber metasurface show a PTE responsivity of 410 V/W at 8 μm radiation. (Nat. Commun. 2023, 14, 3421). Can the author estimate the enhancement of performance if integrated with an artificial structure?
4. The author calculates detectivity using the equation: $D^* = R \cdot (A / 2qV_{\text{dark}})^{1/2}$. This estimation ignores the 1/f noise and g-r noise, which will result in an overestimation for D^* . Please check the recent paper about how to characterize the 2D photodetector. (Nat. Commun. 2023, 14, 2224) I suggest measuring the frequency-dependent NEP and then calculating the D^* , which should be more accurate.
5. The calculated grain size from XRD is smaller than 10 nm for 150-degree PIS sample. However, the HRTEM shows a much larger grain size (>30 nm). The stoichiometric ratio in EDS for PIS treated sample is 76:23, which is far away from the ratio of 2:3 for Bi₂Se₃. Is there any unreacted Bi element residual in the film?

Response to the Reviewers' Comments on Manuscript NCOMMS-23-23409-T

We appreciate the reviewers for considering our manuscript and providing valuable comments. Accordingly, changes have been made in the manuscript, highlighted in yellow. Below is our response to the reviewers' comments.

Response to the Reviewers' comments:

Reviewer 1:

Reviewer's comments:

In this manuscript, the authors prepared metal chalcogenide films on multiple polymeric substrates using pulse irradiation method. Furthermore, the photothermoelectric (PTE) performance can be enhanced due to the thermal coupling between thin films and polymeric substrates. There are several critical issues to be addressed before further consideration.

Response:

We thank the reviewer for the thorough review of our manuscript and appreciate valuable comments that greatly help us to improve the manuscript further.

1. The authors utilize the thermal coupling between the thermoelectric film and underlying substrate to tune the PTE detection performance. Similar strategy can be found in ACS Nano 2013, 7(8), 7271-7277. So, a proper discussion about the link between these two works should be included in the Introduction part.

Response:

We thank you for this important advice. In the work reported by He et al. [ACS Nano 2013, 7, 7271], the scanning photovoltage microscopy (SPVM) results of different substrates (e.g., AlN, glass, and Teflon) show that a substrate with a lower thermal conductivity exhibits a higher PTE response. The relationship between these two works is that the thermal properties of the substrate are associated with the PTE response. However, we believe using thermal conductivity (κ) as the criterion to measure the strength of thermal coupling between the thermoelectric film and underlying substrates may be incomplete. Thermal conductivity indicates how fast the heat propagates according to Fourier's law of heat conduction. In the PTE photodetector, we are concerned about the temperature gradient within the device channel, as heat flux alone is insufficient to describe the temperature

distribution. According to the derivations presented in Supplementary Note 3, we proposed that the thermal diffusivity can give a better description of temperature gradient under localized laser illuminations and, thus, a more appropriate criterion for thermal coupling between the Bi₂Se₃ film and the substrate. Since the thermal diffusivity ($\alpha=\kappa/(\rho C_p)$) can be determined as the thermal conductivity divided by the volumetric heat capacity, the thermal diffusivity shows the ability of a material to conduct thermal energy compared to its ability to store thermal energy. Hence, the thermal diffusivity quantifies the rate at which the temperature smooths out in the material. For a material with a low thermal diffusivity, the heat can hardly move because of its slower heat conduction compared to its volumetric heat capacity, leading to a large temperature gradient. It couples with the thermoelectric film and generates photovoltage via the Seebeck effect. To confirm the validity of our theory, here, the thermal diffusivities of AlN, glass, and Teflon are found to be $1.47 \text{ cm}^2 \text{ s}^{-1}$, $4 \times 10^{-3} \text{ cm}^2 \text{ s}^{-1}$, and $1 \times 10^{-3} \text{ cm}^2 \text{ s}^{-1}$ [<https://www.ioffe.ru/SVA/NSM/Semicond/AlN/thermal.html>; J. Thermal Sci. 2004, 13, 91; Rev. Sci. Instrum. 1996, 67, 3616], respectively, which corresponds well with the SPVM results in that work.

In addition, the substrates studied in the work by He et al. neglected polymeric substrates, which usually possess thermal insulating characteristics. Our work's rapid and low-temperature PIS is readily compatible with those thermally unstable polymeric substrates. This feature enables us to choose materials with suitable thermal properties for diverse application scenarios. In this regard, we have added the corresponding discussion in the Introduction section to connect the thermal coupling effect to the previous work. Changes are also presented below.

On line 22 of page 2 in the revised manuscript:

... In the meantime, a prior work has pointed out that the underlying substrates can also influence the PTE response via thermal effects²². Thus, unveiling the thermal coupling effect between the unexplored polymeric substrates and functional films is essential to improve device performance.

2. For the substrate-supported PTE detector, in addition to the substrate's thermal conductivity, its light absorption ability can also influence the photodetection performance remarkably. To clearly illustrate the contribution related with the substrate, the absorption spectra in the visible-near

infrared regions of metal chalcogenide films and polymeric substrates (PDMS, PET, PEN, and PI) should be provided.

Response:

We appreciate the reviewer for this constructive suggestion. The absorption spectra from visible to near-infrared of Bi₂Se₃ and polymeric substrates (i.e., PDMS, PET, PEN, and PI) are provided in Supplementary Fig. 13. It is observed that all flexible substrates show poor absorption with minor differences in absorption spectra. In contrast, the Bi₂Se₃ film demonstrates the higher absorption. This observation indicates that different responsivities in Fig. 3g can be ascribed to the inherent thermal properties of the underlying substrate itself instead of its optical absorption. We have added corresponding discussions to address this concern.

On line 30 of page 4 in the revised manuscript:

... It is also noted that all flexible substrates show poor absorption with minor differences in absorption spectra. In contrast, the Bi₂Se₃ film demonstrates higher absorption (Supplementary Fig. 13), further underscoring that the device performance is primarily governed by the inherent thermal properties of the underlying substrate itself instead of its optical absorption. ...

On page 17 in the revised Supplementary Information:

Supplementary Figure 13 | Optical absorption spectra of the PIS Bi₂Se₃ film and the four substrates utilized in this work.

3. In Figure 4b, the authors provided a scanning photovoltage mapping of Bi₂Se₃ PTE detector (Incident wavelength: 532 nm; Power: 0.5 mW; Spot diameter: 3 μm). The maximum photovoltage response is about 50 μV. As a result, the calculated responsivity is just 0.1 V/W, which obviously deviate from the claimed responsivity in Figure 4h (52.4 V/W at 532 nm). It is necessary to check the data consistency and accuracy.

Response:

We appreciate the reviewer for pointing out this important issue. We thoroughly re-examined the experimental setup. We found that the recorded light power data was derived directly from the laser output, while the actual laser energy applied to the device underwent noticeable attenuation. Here, we used a calibrated light power meter to measure the actual power, yielding a value of 1.3 μW. With this calibrated value, the calculated responsivity is ~ 38.5 V W⁻¹, which closely approximates the claimed value. Importantly, we wish to highlight that the homemade testing system for the SPVM measurement is more inclined to provide qualitative results. As such, we believe that the measured errors are well within a range that can explain the mechanism of the PTE effect. The corrected light power is revised as shown below:

On line 38 of page 7 in the revised manuscript:

... A high spatial resolution of the V_{ph} was obtained by a 532 nm laser with a focused spot (diameter of 3 μm) and a constant power (1.3 μW). ...

4. In Line 8 of Page 4, the authors claimed that the employment of PIS technique can avoid the Se sublimation. However, the EDX data in Supplementary Fig. 8f show that the element composition after PIS and CFA treatment is Bi₂Se_{0.6} and Bi₂Se_{2.5}, respectively. Please explain this inconsistency.

Response:

We thank the reviewer for this critical comment. We agree that the EDS data are inconsistent with the manuscript's descriptions. Hence, we checked the source data and found that incorrect data was used for the PIS-treated sample. Thus, we conducted the EDS test again and corrected Supplementary Fig. 8. The corrected data indicates that the element composition of the film is close

to Bi_2Se_3 , indicating the effective prevention of Se loss by PIS.

On page 11 in the revised Supplementary Information:

Supplementary Figure 8 | SEM images indicate that prolonged annealing destroys the film quality. a,b, Low-magnification (a) and high-resolution (b) SEM images of the Bi_2Se_3 film after PIS. **c-e,** Larger SEM image (c) showing the degradation of the Bi_2Se_3 film after CFA, and detailed morphologies under low-magnification (d) and high-resolution (e). **f,g,** EDS spectra and the determination of the element composition after PIS (f) and CFA (g).

5. The formula of specific detectivity (D^*) is not correct (Line 6, Page 7). Usually, the D^* is calculated using the following formula:

$$D^* = \sqrt{A} / NEP$$

$$NEP = \sqrt{4kBT} / RV$$

where kB is the Boltzmann constant, T is the absolute temperature, and R is the device resistance.

Response:

We are grateful for this valuable comment and agree that we have used the inappropriate formula for D^* evaluation. The formula for the noise equivalent power ($NEP = \sqrt{4k_b TR/R_v}$) only considers the thermal noise [Nat. Commun. 2023, 14, 2224]. Other noises, such as shot noise, g-r noise, and $1/f$ noise, are neglected. Despite the significantly suppressed shot noise because the device operates under zero bias [ACS Nano 2019, 13, 13285], the detectivity will still be overestimated. Specifically, the formula proposed by the reviewer yields a NEP value of 4.12×10^{-11} W Hz^{-1/2}, and the corresponding D^* value is 1.88×10^8 Jones. To determine the correct D^* , we measured the spectral density of the voltage noise and calculated the NEP value via $NEP = v_n/R_v$, where v_n is the voltage noise, and R_v is the responsivity. This method has been widely used to determine the D^* of PTE detectors [Nat. Commun. 2022, 13, 4560; Nano Lett. 2022, 22, 5929; Nat. Commun. 2023, 14, 3421; Nano Lett. 2023, 23, 812]. The D^* exhibits a value of 1.85×10^7 Jones at 1550 nm, comparable with those reported in earlier studies on PTE detectors. However, this value is smaller than the D^* obtained using the previous method, indicating that the previous approach does not fully consider the impact of noises. To address this issue, we provided the voltage noise spectrum, recalculated the detectivity, and revised the corresponding figures and statements. All changes are listed below:

On line 34 of page 5 in the revised manuscript:

... The power-dependent detectivity (D^*) extracted from the noise equivalent power (NEP) reaches a maximum value of 1.85×10^7 Jones at 1550 nm (Supplementary Fig. 22 and 23), comparable with recently reported PdSe₂ and MoTe₂ PTE photodetectors⁴⁴⁻⁴⁶.

On line 25 of page 7 in the revised manuscript:

The detectivity (D^*) was extracted as:

$$D^* = \frac{\sqrt{A\Delta f}}{NEP} \quad (3)$$

where A is the device area, and Δf is the bandwidth. The NEP was calculated by $NEP = v_n/R$, where v_n is the measured voltage noise (Supplementary Fig. 22), and R is the responsivity. ...

On page 27 in the revised Supplementary Information:

Supplementary Figure 22 | Spectral density of the voltage noise under zero bias.

Supplementary Figure 23 | Plot of detectivity D^* as a function of the incident IR laser power.

On page 32 in the revised Supplementary Information:

Supplementary Figure 18 | Performance at the visible range. a, Compiled responsivity R as a function of the laser power at the visible range. b, Plot of detectivity D^* as a function of the incident wavelength.

6. In Line 33 of Page 5 and Supplementary Figure 21, the authors mentioned the "photon thermal transport". What's the meaning of photon thermal transport?

Response:

We thank the reviewer for pointing out this valuable issue. In our work, we found that the response time increases when a bias is applied to the device, agreeing well with the PdSe₂-based PTE photodetector [ACS Nano 2022, 1, 295]. Such extension of response time can be ascribed to the photoconductive effect (PCE), which dominates over the PTE effect under external bias. Indeed, the PCE of Bi₂Se₃ has been reported for near-infrared detection [Appl. Surf. Sci. 2014, 316, 341; Sci. Rep. 2016, 6, 19138; Adv. Funct. Mater. 2018, 28, 1802707]; however, its PTE effect has evidently been overlooked.

To exclude the PTE effect under bias, we moved the incident light spot to the center of the device channel. At this position, the temperature gradient is absent between two electrodes (Fig. 4a(iii)); thus, there is no electric potential difference between the electrodes. Under this condition, the photocurrent is generated via PCE under 1 V bias (Supplementary Fig. 24c). Different from the PTE effect, the PCE stems from the light-induced carrier density modification and its relatively longer rise and decay times primarily result from carrier scattering and electronic trapping [Adv. Mater. 2018, 30, 1804332; ACS Photonics 2017, 4, 2335; Adv. Funct. Mater. 2018, 28, 1706437]. To clarify this point, we removed the statement regarding photon thermal transport from the text, added

relevant discussions, and revised corresponding figures.

On line 42 of page 5 in the revised manuscript:

... Under external bias, the photoconductive effect starts to contribute to the response (Supplementary Fig. 24c and d), and carrier scattering and electronic trapping are responsible for the relatively longer response time. ...

On page 28 in the revised Supplementary Information:

Supplementary Figure 24 | Time-resolved photoresponse of the PTE detector. a, Response time of the device under zero bias. **b,** Frequency response showing a 3 dB frequency cutoff at ~ 20 Hz. **c,** Time-resolved response with an external bias of 1 V when the light spot is positioned at the center of the device channel. **d,** Artist's rendition of the contribution of PTE and the photoconductive effect (PCE) under zero bias and 1 V bias.

7. The authors attributed the fast response (around 50 ms) to the hot-carrier diffusion. However, this time is quite typical for thermal detectors where the phonon dominates the heat transport. For

example, the response time of commercial infrared thermopile is usually around 20 ms. On the other side, the generated hot carriers approach thermal equilibrium with phonons at picosecond time scale, which also cannot support the author's conclusion.

Response:

We thank the reviewer for pointing out this important issue and providing valuable suggestions. The response time in the range of milliseconds originates from the phonon thermal transport, not the hot-carrier diffusion. The response time of ~ 50 ms cannot support our current conclusion. We sincerely apologize for this mistake and have made revisions to the relevant statement in the revised manuscript and Supplementary Information, with changes presented below:

On line 40 of page 5 in the revised manuscript:

... The scale of the PTE response time reflects the phonon-dominated transport in the Bi₂Se₃ film and the thermal equilibrium rate that largely depends on the thermal conductivity and the heat capacity of materials⁴³. ...

8. As a flexible photodetector, the bending or stretching stability is one of the important appraisal aspects. It would be better to provide the corresponding results.

Response:

We appreciate the reviewer's valuable suggestion. We tested the bending performance of the PTE photodetector on the PI substrate. As shown in Supplementary Fig. 24, the time-dependent photoresponse curves still exhibited reproducible characteristics without obvious decay of responsivity (< 5 %) after 5000 bending cycles. Corresponding discussions and figures are added, with changes presented below:

On line 3 of page 6 in the revised manuscript:

... Also, the device maintains excellent reproducible photoresponse characteristics, with no discernable degradation of responsivity (~ 5 %) after 5000 bending cycles (Supplementary Fig. 26).

On page 30 in the revised Supplementary Information:

Supplementary Figure 26 | Bending stability of the PTE photodetector. a, Photovoltage and responsivity after different bending cycles. **b,** Photoresponse curves before and after the bending test. The bending radius is 5 mm.

9. In the Abstract part, the wavelength for the 71.9 V/W responsivity should be provided.

Response:

We thank the reviewer for the reminder. We have added the wavelength (1550 nm) for the responsivity in the Abstract section of the revised manuscript.

On line 37 of page 1 in the revised manuscript:

... ,resulting in a responsivity of 71.9 V/W and a response time of less than 50 ms at 1550 nm, surpassing most of its counterparts. ...

10. Some small mistakes should be corrected:

- 1) Line 8, Page 2, "...processing methods that enables direct low-temperature deposition...".
- 2) Line 27, Page 3, "The calculated adiabatic temperature is ~ 722 °C, which is much lower than the melting temperature of both Bi (271.4 °C) and Se (220.8 °C)".

Response:

We thank for the reviewer's kind reminder. We have revised the relevant part in the revised manuscript following the reviewer's suggestion.

On line 9 of page 2 in the revised manuscript:

... processing methods that enable direct low-temperature deposition ...

On line 29 of page 3 in the revised manuscript:

... and the calculated adiabatic temperature is ~ 722 °C, which is much higher than the melting temperatures of both Bi (271.4 °C) and Se (220.8 °C), ...

Reviewer 2:

Reviewer's comments:

The authors have demonstrated a pulse radiation method to synthesize thermoelectric thin films. The low growth temperature requirement is quite suitable for flexible optoelectronic devices, directly growing on polymer substrates without degradation. Moreover, the low thermal diffusivities of these polymer substrates contribute to much enhanced PTE performance. This work is interesting and comprehensive, which should draw the attention of the researchers working in 2D materials synthesis, photodetectors, and thermoelectric materials. However, I have several questions about the measurement and find several problems in the manuscript, therefore this work should be reconsidered after answering the following questions:

Response:

We thank you for the thorough review of our work and appreciate constructive suggestions that help us to improve the quality and depth of our work.

1. The authors use the thermal camera to record the temperature difference under laser radiation. However, I think this measurement will not be accurate for small dimension devices, considering that the channel length is around 60 μm and the focus beam size is smaller than 3 μm . The temperature reading from the software should be the average of several pixel signals, and I doubt that the pixel resolution of a thermal camera can reach so high. I think the real temperature change induced by the laser should be even higher.

Response:

We appreciate the reviewer pointing out this issue, which we did not describe clearly. To ensure the infrared camera's accuracy, we utilized a larger beam size to acquire the data. As a result, the diameter of the light spot detected by the thermal camera is approximately 1.5 mm. According to the specifications of the thermal camera used in this work (Fluke Ti480 PRO), the spatial resolution is 0.98 mRad, the detector resolution is 640×480 , and the field of view is $34^\circ\text{H} \times 24^\circ\text{V}$ [<https://www.fluke.com/en-us/product/thermal-cameras/ti480-pro>]. At a typical detection distance (15 cm), the resolution is $0.98 \text{ mRad} \times 0.015 \text{ m} = 0.14 \text{ mm}$, larger than the spot size. Besides, the number of pixels occupied by the light spot in the detector is approximately 10 ($1.5 \text{ mm} \times 360^\circ \times 640 / (15 \text{ cm} \times (2\pi) \times 34^\circ) \approx 10$). Hence, the temperature reading from the infrared camera is

reliable. In order to address the reviewer's concern, we added details about the data acquisition by the infrared camera in the Methods section, with changes presented below:

On line 32 of page 7 in the revised manuscript:

... Specifically, this test was carried out with a larger beam size (~ 1.5 mm) to ensure it exceeded the minimum resolution of the infrared camera. Moreover, the measurement was synchronized with ...

2. The Seebeck coefficient is derived from Fig. 4b. The author didn't provide information about how they got this figure. If the temperature change is also read from the thermal camera, then I will also question its accuracy. It is important to compare the Seebeck coefficients for PIS Bi₂Se₃ thin film with that synthesized with conventional methods. To get an absolute Seebeck coefficient, the authors can follow the methods in Nat. Commun. 2023, 14, 1938; Nano Letter. 2014, 14, 5, 2730.

Response:

We thank the reviewer pointing out this concern and are sorry for the unclear description of the method. It should be noted that the Seebeck coefficient of the scalable Bi₂Se₃ film in this work was obtained from a specialized thermoelectric material test system (CTA-3, Beijing Cryoall Science and Technology Co., Ltd.). The schematic of the equipment is shown in Fig. R1, where the sample is fixed onto a hot end (T_1) and a cold end (T_2) via a sample holder. A heater in the hot end creates a temperature gradient. Two thermocouples (T_3 and T_4) contact the film to measure the temperatures and potentials with their Pt electrodes acting as conductors. Once the temperature difference stabilizes, we can harvest the temperature difference (ΔT) and the potential difference (ΔU) simultaneously. The data for the Seebeck coefficient can be compiled by the software for the ΔT - ΔU curve.

Figure R1 | Schematic of the configuration in the Seebeck coefficient test.

In fact, the methods for obtaining the Seebeck coefficient mentioned by the reviewer in this comment are similar to our work, where a heater creates a temperature gradient, and two thermocouples measure the temperature. The only difference is that they utilized photolithography to fabricate the test configuration due to the limited active areas of their materials. The PIS in our work is compatible with scalable thin film deposition techniques. Thus, the conventional bulk/film thermal property testing methods are applicable. In the meantime, we agree that the comparison with Bi_2Se_3 synthesized by conventional methods is important in this work. Therefore, we compared the Seebeck coefficient of Bi_2Se_3 with previous works. However, it can be seen from Supplementary Fig. 8 that the film's integrity is largely disrupted after CFA, making it difficult to conduct the thermoelectric measurements.

We have added details of Seebeck coefficient measurement in the Methods section and comparisons of Seebeck coefficients of Bi_2Se_3 with previous works. The changes are presented below:

On line 11 of page 5 in the revised manuscript:

... comparable with other Bi_2Se_3 films synthesized with conventional methods (Supplementary Table 2) and other chalcogenide counterparts³⁴⁻³⁶, ...

On line 11 of page 7 in the revised manuscript:

.. The Seebeck coefficient was obtained from a thermoelectric material test system (CTA-3, Cryoall). Before measurement, two thermocouples contacted the film to obtain U-I curves at room temperature. The test proceeded when the coefficient of determination for the fitted U-I curve exceeded 0.99999, which indicated good contact of the thermocouple with the surface. A heater was employed to create a temperature gradient in the film. Once the temperature difference stabilized, temperatures at the two thermocouple locations and the electrical potential between them were measured simultaneously. Then, the Seebeck coefficient was extracted from the ΔT - ΔU curve.

On line 20 of page 20 in the revised supplementary information:

Supplementary Table 2 | Comparison of the Seebeck coefficients of Bi₂Se₃ films synthesized by conventional methods.

Method	Seebeck coefficient ($\mu\text{V K}^{-1}$)	Year	Ref.
Hydrothermal method	-78	2010	20
Hydrothermal method	-113	2010	21
CVD	-99.9	2021	22
Molecular beam epitaxy	-102.8	2022	23
Magnetron sputtering	-153	2022	24
PIS	-111	2023	This work

3. The author claims to get a record PTE responsivity (71.9 V/W). However, the recent published Te nanowire on perfect absorber metasurface show a PTE responsivity of 410 V/W at 8 μm radiation. (Nat. Commun. 2023, 14, 3421). Can the author estimate the enhancement of performance if integrated with an artificial structure?

Response:

We are grateful for this constructive advice. Metasurfaces have been utilized as plasmonic absorbers to enhance the photoresponse or empower the PTE detector with a polarization-resolved detection capability [ACS Photonics 2016, 3, 936; Nat. Commun. 2018, 9, 5190; Light: Sci. Appl. 2020, 9, 126; Nat. Commun. 2022, 13, 4560; ACS Nano 2022, 16, 17263]. It can be inspirational for readers in the combination of thermal coupling and surface plasmonic absorption. Here, we demonstrated

that the device's response was enhanced when integrated with plasmonic meta-atoms (Supplementary Fig. 14), indicating the potential of metasurfaces for PTE devices.

On line 34 of page 4 in the revised manuscript:

... In addition, surface meta-materials can be employed as plasmonic absorbers to improve performance³². It is observed that the photoresponse increases by ~ 60 % after the integration with gold-based meta-materials (Supplementary Fig. 14). This result may prompt further investigations into the synergistic effect of thermal coupling and surface plasmonic absorption. ...

On page 18 in the revised supplementary information:

Supplementary Figure 14 | Performance improvement of PTE detectors through meta-material integration. **a**, Optical image of the Bi₂Se₃ film photodetector integrated with a gold microstructure array. **b**, Temporal response curves with and without integration with meta-materials. **c**, Performance improvement with and without meta-materials.

4. The author calculates detectivity using the equation: $D^* = R^* / (A/2qV_{dark})^{1/2}$. This estimation ignores the $1/f$ noise and $g-r$ noise, which will result in an overestimation for D^* . Please check the recent paper about how to characterize the 2D photodetector. (Nat. Commun. 2023, 14, 2224) I suggest measuring the frequency-dependent NEP and then calculating the D^* , which should be more accurate.

Response:

We thank the reviewer for pointing out this issue, which helps improve our manuscript's quality. The current formula for the estimation of detectivity neglects $1/f$ noise and $g-r$ noise. Hence, we followed the method as suggested by the reviewer and provided the voltage noise spectrum to calculate NEP and D^* . After the correction, the D^* exhibits a value of 1.85×10^7 Jones at 1550 nm,

comparable with those reported in earlier studies on PTE detectors. In order to address this concern, corresponding figures and statements have been revised, and all changes are listed below:

On line 34 of page 5 in the revised manuscript:

... The power-dependent detectivity (D^*) extracted from the noise equivalent power (NEP) reaches a maximum value of 1.85×10^7 Jones at 1550 nm (Supplementary Fig. 22 and 23), comparable with recently reported PdSe₂ and MoTe₂ PTE photodetectors⁴⁴⁻⁴⁶.

On line 25 of page 7 in the revised manuscript:

The detectivity (D^*) was extracted as:

$$D^* = \frac{\sqrt{A\Delta f}}{NEP} \quad (3)$$

where A is the device area, and Δf is the bandwidth. The NEP was calculated by $NEP = v_n/R$, where v_n is the measured voltage noise (Supplementary Fig ...), and R is the responsivity. ...

On page 27 in the revised Supplementary Information:

Supplementary Figure 22 | Spectral density of the voltage noise under zero bias.

Supplementary Figure 23 | Plot of detectivity D^* as a function of the incident IR laser power.

On page 32 in the revised Supplementary Information:

Supplementary Figure 28 | Performance at the visible range. a, Compiled responsivity R as a function of the laser power at the visible range. **b,** Plot of detectivity D^* as a function of the incident wavelength.

5. The calculated grain size from XRD is smaller than 10 nm for 150-degree PIS sample. However, the HRTEM shows a much larger grain size (>30 nm). The stoichiometric ratio in EDS for PIS treated sample is 76:23, which is far away from the ratio of 2:3 for Bi_2Se_3 . Is there any unreacted Bi element residual in the film?

Response:

We are grateful for the reviewer's valuable comments. We have shown that the PIS Bi_2Se_3 crystallizes even when the synthesis temperature is as low as 150 °C with a grain size of ~ 10 nm. From GIWAXS patterns, the prime crystallite extends along the substrate surface, and the grain sizes increase with the PIS temperature. In specific, the HRTEM images in Fig. 2 and Supplementary Fig. 5 are obtained from the 300-degree PIS, showing a larger grain size (> 30 nm). According to Supplementary Fig. 2d, the calculated grain size of the Bi_2Se_3 film with the 300-degree PIS is close to 30 nm, consistent with the HRTEM images. In the meantime, we found that incorrect EDS data was used, which is far from the stoichiometric ratio of Bi_2Se_3 . Hence, we conducted a new EDS test and revised the corresponding figures. The corrected data indicates that the element composition of the film is close to Bi_2Se_3 , indicating the effective prevention of Se loss by PIS and no excessive Bi in the film.

On page 13 in the revised manuscript:

Fig. 2 | PIS synthesis of large-area PTE film and characterization. a-d, GIWAXS patterns of Bi_2Se_3 films after PIS at 150 (a), 200 (b), 250 (c), and 300 °C (d) for 1s, respectively. e,f, HRTEM

image and SAED pattern of the film after evaporation. **g,h**, HRTEM image, and SAED pattern of the film after the **300 °C PIS**. **i,j**, Typical snapshots taken in the AIMD simulation during the interdiffusion of Se and Bi layers (**g**) and attachments of Se and Bi atoms to the edge of the primary Bi_2Se_3 cluster (**h**), almost completing two columns of atoms on both sides (highlighted with colored boxes).

On page 11 in the revised Supplementary Information:

Supplementary Figure 8 | SEM images indicate that prolonged annealing destroys the film quality. a,b, Low-magnification (**a**) and high-resolution (**b**) SEM images of the Bi_2Se_3 film after PIS. **c-e**, Larger SEM image (**c**) showing the degradation of the Bi_2Se_3 film after CFA and detailed morphologies under low-magnification (**d**) and high-resolution (**e**). **f,g**, EDS spectra and the determination of the element composition after PIS (**f**) and CFA (**g**).

List of other changes:

1. The units of diffusivity in Fig. 3g and Supplementary Table 1 are revised to be consistent with the main text.

On page 14 in the revised manuscript:

Fig. 1 | Thermal management of polymeric substrates. a,b, Thermal simulations on SiO₂ (a) and PI (b) surfaces based on the device architecture. S and D denote source and drain electrodes, respectively. **c,** Optical image of the Bi₂Se₃ film photodetector. **d,e,** Corresponding temperature (d) and gradient (e) profiles across the device channel. Two vertical dashed lines indicate the interfaces between electrodes and channels. **f,** Temporal response curves comparing the PTE film on Si/SiO₂ and PI substrates. **g,** Responsivity on four typical flexible substrates (PDMS, PET, PEN, and PI) and summary of the corresponding diffusivity (Supplementary Table 1), indicating an inverse correlation between them.

On page 18 in the revised supplementary information:

Supplementary Table 1 | Diffusivity of four typical substrates used in this work.

Substrate	Thermal diffusivity ($\text{mm}^2 \text{s}^{-1}$)	Ref.
PI	0.21	15
PEN	0.15	16
PET	0.16	16
PDMS	0.07	17

2. Several minor mistakes were corrected:

On line 21 of page 2 in the revised manuscript:

... Previous studies have reported that the responsivity of PTE detectors can be improved by utilizing electrical gating, surface plasmonics, antenna coupling, and phonon **absorption**, which result in a higher temperature gradient or enhanced Seebeck coefficient¹⁷⁻²¹. ...

On line 29 of page 3 in the revised manuscript:

... In this work, the exothermic reaction of $2\text{Bi} + 3\text{Se} \rightarrow \text{Bi}_2\text{Se}_3$ exhibits a negative formation enthalpy ($\sim -140 \text{ kJ mol}^{-1}$), and the calculated adiabatic temperature is $\sim 722 \text{ }^\circ\text{C}$, which is much **higher** than the melting **temperatures** of both Bi ($271.4 \text{ }^\circ\text{C}$) and Se ($220.8 \text{ }^\circ\text{C}$), ...

3. Equation numbers in Supplementary Note 3 were corrected.

4. The statements of the data availability and author contributions have been added to the revised manuscript.

REVIEWER COMMENTS

Reviewer #1 (Remarks to the Author):

The authors have addressed most of the comments. However, I still have concerns about the accuracy, reliability and innovation of this paper.

1. How is the laser power (532 nm) attenuated from 0.5 mW (incident power) to 1.3 uW (actual power on the sample)? The accurate measurement of laser power is extremely important for sensitivity calculation (in this case, the sensitivity changes from 0.1 V/W to 38.5 V/W just due to the inaccuracy).
2. How does the EDS ratio of Bi/Se change from 72/23 (previous version) to 39/60 (revised version), while the EDS intensities look inconsistent (SI Fig.8) ?
3. If the detectivity decreases from 2.84×10^9 Jones to 1.85×10^7 Jones after correction, such performance is much worse than the previous report (1.07×10^8 Jones, MoTe₂, Ref. 46).

Reviewer #2 (Remarks to the Author):

The authors have provided a comprehensive study for their low-temperature synthesized thermoelectric metal chalcogenide for PTE photodetector. Most of my concerns have been addressed in the revised manuscript. I still have one question about the broadband response. The photo-response decreases when the wavelength changes from 400-635 nm, which follows the trend of the absorption spectra. However, why does the response increase while the absorption is relatively low at 1550 nm. Overall, the manuscript is of high quality and can be published in Nature Communications.

Response to the Reviewers' Comments on Manuscript NCOMMS-23-23409A

We appreciate the reviewers for considering our manuscript and providing valuable comments. Accordingly, changes have been made in the manuscript, highlighted in yellow. Below is our response to the reviewers' comments.

Reviewer #1 (Remarks to the author):

The authors have addressed most of the comments. However, I still have concerns about the accuracy, reliability and innovation of this paper.

Reply:

We thank the reviewer for the thorough review of our manuscript and constructive comments on our work.

1. How is the laser power (532 nm) attenuated from 0.5 mW (incident power) to 1.3 μ W (actual power on the sample)? The accurate measurement of laser power is extremely important for sensitivity calculation (in this case, the sensitivity changes from 0.1 V/W to 38.5 V/W just due to the inaccuracy).

Reply:

We appreciate the reviewer for bringing up this matter, which was not adequately explained in our initial description. Initially, we measured the power of the laser directly emitted from the laser source (0.5 mW). After revision, we measured the actual power of the laser that directly illuminates the device (1.3 μ W). Here, the laser emitted from the laser source inevitably experiences attenuation in intensity as it passes through our homemade optical paths and laser focusing systems. Thus, our measurement of the actual laser power ensures the accuracy of the responsivity.

2. How does the EDS ratio of Bi/Se change from 72/23 (previous version) to 39/60 (revised version), while the EDS intensities look inconsistent (SI Fig.8)?

Reply:

We thank the reviewer for this question. After rechecking the EDS data, we found that the labeling of the EDS peak around 1.75 keV was incorrect; this peak is attributed to the Si element instead of the Bi element. Hence, the inconsistency stems from the Si EDS peak, which does not affect the ratio of Bi/Se in elemental analysis. To avoid misinterpretation, we have corrected the label and provided the EDS spectra with the full energy range in the revised Supplementary Information. We are sorry for this mistake.

On page 11 of the revised Supplementary Information:

Supplementary Figure 1 | SEM images indicate that prolonged annealing destroys the film quality. a,b, Low-magnification (a) and high-resolution (b) SEM images of the Bi_2Se_3 film after PIS. **c-e**, Larger SEM image (c) showing the degradation of the Bi_2Se_3 film after CFA and detailed morphologies under low-magnification (d) and high-resolution (e). **f,g**, EDS spectra and the determination of the element composition after PIS (f) and CFA (g).

3. If the detectivity decreases from 2.84×10^9 Jones to 1.85×10^7 Jones after correction, such performance is much worse than the previous report (1.07×10^8 Jones, MoTe_2 , Ref. 46).

Reply:

We thank the reviewer for this critical comment. In contrast to the single crystal MoTe_2 with the higher detectivity achieved by mechanical exfoliation (Ref. 46), our fabrication strategy offers scalability due to the thin film deposition technique combined with low-temperature PIS. This feature aligns with the requirement of batch production in thin film micromanufacturing. We also want to note that the revised detectivity in the order of 10^7 Jones is still comparable with previous works [ACS Nano 2022, 16, 295; Adv. Funct. Mater. 2021, 31, 2104787; Nat. Commun. 2022, 13, 1835]. In order to maintain the manuscript's accuracy, we revised the relevant reference concerning the detectivity.

On line 36 of page 5 in the revised manuscript:

... ,comparable with some recently reported PTE photodetectors^{44,45}.

Reviewer #2 (Remarks to the author):

The authors have provided a comprehensive study for their low-temperature synthesized thermoelectric metal chalcogenide for PTE photodetector. Most of my concerns have been addressed in the revised manuscript.

Reply:

We thank you for the thorough review of our work and appreciate your positive comments.

1. I still have one question about the broadband response. The photo-response decreases when the wavelength changes from 400-635 nm, which follows the trend of the absorption spectra. However, why does the response increase while the absorption is relatively low at 1550 nm.

Reply:

We thank the reviewer for pointing out this issue, which we did not describe clearly. The absorption spectrum of Bi₂Se₃ proves the independence of its absorption from the photoresponse. It is the inherent thermal properties of the underlying substrates that affect the responsivity (Fig. 3 and Supplementary Note 3). In the meantime, it should be noted that the response increase at 1550 nm is attributed to the more pronounced heat effect of near-infrared light. Hence, the temperature changes in the substrate are more sensitive in the near-infrared range, leading to a higher responsivity due to the enhanced thermal coupling between the Bi₂Se₃ and the substrate.

Overall, the manuscript is of high quality and can be published in Nature Communications.

Reply:

We thank you again for the valuable comments.

List of other changes:

1. The heading of “CONCLUSIONS” has been changed to “**DISCUSSION**”.

REVIEWER COMMENTS

Reviewer #1 (Remarks to the Author):

There are too many mistakes in this manuscript from the beginning, which make the results unreliable and unacceptable. I strongly suggest the authors to double-check and discuss all these mistakes with your group members before submitting to any journals.

Reviewer #2 (Remarks to the Author):

My previous concerns are all addressed. Nonetheless, the author's response is not sufficient for the questions raised by reviewer 1. While the primary innovation lies in establishing a distinctive growth method for thermoelectric films on polymer substrates, the optoelectrical performance of the device is equally significant. The authors should provide more details to substantiate the reliability and noteworthy of the results.

1. It is strange that the laser power decrease so much, which is more than two orders in the testing system. Normally, in a photocurrent mapping system, the power loss is smaller than one order. Can the author provide the information for testing setup and explain the reason for the loss?
2. The EDS result should come from samples grown on polymer, then where does the Si element come from? By the way, the XPS should be a more precise technique for element composition analysis. The author can even do Ar⁺-etched XPS to check the composition at different thickness.
3. The detectivity can be extremely high for 2D based detectors in visible and NIR range ($>1E10$ Jones). The performance shown here is just comparable with PTE type detector. The author should explain the advantage of the Bi₂Se₃ film produced by this technology compared with non-PTE type materials and also possible solutions to enhanced its performance.

Response to the Reviewers' Comments on Manuscript NCOMMS-23-23409B

We appreciate the reviewers for considering our manuscript and providing valuable comments. Accordingly, changes have been made in the manuscript, highlighted in yellow. Below is our response to the reviewers' comments.

Reviewer #1 (Remarks to the author):

There are too many mistakes in this manuscript from the beginning, which make the results unreliable and unacceptable. I strongly suggest the authors to double-check and discuss all these mistakes with your group members before submitting to any journals.

Reply:

We are grateful for the valuable comments previously provided by the reviewer on our work, which greatly helped us to improve the manuscript. Meanwhile, we sincerely apologize for not being able to provide a satisfactory response, and we have provided more results and discussions to substantiate the reliability and noteworthy of this work, aiming to eliminate any inaccuracies to meet the required standards.

Specifically, we have provided more discussions here in response to each comment raised by the reviewers, which are summarized as follows:

1. A more detailed description of our scanning photovoltage mapping (SPVM) measurement is provided. After inspecting each optical component in the external optical path, we found that the main power loss comes from improper fiber coupling. Despite the observed power loss or attenuation along the optical pathway, the power of light illuminating the device is definite and remains constant throughout the measurement. Finally, we performed a new SPVM test with a dedicated optoelectronic scanning system to validate the reliability of SPVM results, and the obtained responsivity is in good agreement with the previously claimed responsivity.
2. The XPS depth profile analysis was employed to characterize the film's composition accurately. After comparing the compositional depth profile results of the Bi₂Se₃ films produced by PIS and CFA, it again indicates that the PIS effectively prevents the loss of Se in the film, agreeing well with our previous statements.
3. The advantages of the PIS-produced Bi₂Se₃ film are highlighted in comparison with other PTE detectors and non-PTE detectors: i) our wafer-scale pulse irradiation synthesis (PIS) achieves an ultra-fast processing speed (< 1 s) under a low temperature (~ 150 °C), unachievable by other conventional fabrication techniques; ii) a unique thermal coupling mechanism is firstly proposed and used for further manipulating the performance of PTE devices; iii) the PTE photodetector based on Bi₂Se₃ realizes a wide spectral detection from visible to infrared optical communication band, which is not easily achievable by non-PTE detectors limited by their semiconductor bandgaps. Overall, our innovative PIS synthesis technique and the newly proposed thermal coupling mechanism hold great potential in advancing large-area, low-cost, flexible optoelectronic devices.

We sincerely hope the reviewer can refer to our more detailed responses provided to reviewer #2 for further information.

Reviewer #2 (Remarks to the author):

My previous concerns are all addressed. Nonetheless, the author's response is not sufficient for the questions raised by reviewer 1. While the primary innovation lies in establishing a distinctive growth method for thermoelectric films on polymer substrates, the optoelectrical performance of the device is equally significant. The authors should provide more details to substantiate the reliability and noteworthy of the results.

Reply:

We sincerely appreciate the reviewer for highlighting the shortcomings in our previous revision. We have diligently conducted additional experiments and thoroughly revised the manuscript to ensure its accuracy and reliability. We understand the reviewer's concerns and are committed to addressing them to the best of our abilities.

1. It is strange that the laser power decreases so much, which is more than two orders in the testing system. Normally, in a photocurrent mapping system, the power loss is smaller than one order. Can the author provide the information for testing setup and explain the reason for the loss?

Reply:

We are grateful to the reviewer for this critical comment, and we also noticed a significant power loss from the laser output to the actual power applied to the device. Different from specialized photocurrent mapping systems, where highly integrated optical paths can effectively reduce light power loss, our laser passes through an external optical path and couples into an optical fiber before entering our homemade optoelectronic scanning system. Notably, we identified a substantial power loss in the fiber coupling, with the light intensity being reduced by approximately one order of magnitude. Since the requirements for various parameters in such optic coupling are stringent (e.g., high-quality fiber end face, properly focused beam with no aberrations and proper mechanical positioning of the fiber end face), our inadequate optical coupling is likely responsible for this power loss. Together with the attenuation caused by the focusing system and other optical components (e.g., optical lenses, reflectors, and attenuators), the light intensity reaching the device is ultimately decreased by more than two orders of magnitude. It is important to highlight that, despite the observed light loss or attenuation along the pathway, the power of light illuminating the device is definite and remains constant throughout the measurement. Thus, we believe this power loss does not impact the device's photovoltage distribution evaluation.

Here, to further validate the reliability of scanning photovoltage mapping (SPVM) results, we conducted a new test using a dedicated high-resolution optoelectronic scanning system (ScanPro Advance, Metatest). Before scanning, a light power meter was calibrated and placed directly below the sample stage to measure the precise light power applied to the device (Fig. R1a and b). We can read the light power value as 3.4 μW (Fig. R1c). According to the retested SPVM data, where the photovoltage under zero bias is about 180 μV (Fig. 4c), the calculated responsivity is 52.9 V/W, consistent with the claimed responsivity at 532 nm (Fig. S28). Based on the retested results, we

have replaced all corresponding figures and modified relevant descriptions in the manuscript.

Figure R1 | a,b, Laser power measurement directly applied to the device. c, Light power meter reading from the software.

On line 6 of page 5 in the revised manuscript:

... The scanning photovoltage mapping (SPVM) directly exhibits an opposite photovoltage (V_{ph}) of $\pm 180 \mu V$ at two ends of the channel (Fig. 4b). ...

On line 10 of page 8 in the revised manuscript:

SPVM measurement. To investigate the working mechanism of the PTE detector, we characterized the photovoltage mapping by an integrated optoelectronic scanning system (ScanPro Advance, Metatest). The spot size is controlled to be $3 \mu m$ to ensure a high spatial resolution of the mapping. Before scanning, a light power meter was placed directly below the sample to test the actual light power applied to the device ($3.4 \mu W$). The photovoltage was monitored by a source/measure unit (2636B, Keithley).

On page 16 of the revised manuscript:

Fig. 1 | Mechanism and performance of PTE photodetectors. **a**, Schematic of the local laser-induced temperature distribution T and potential distribution ΔV along the channel when the laser spot moves from the left electrode to the right one. The white arrow indicates the diffusion direction of hot electrons. S , T_l , and T_r refer to the Seebeck coefficient and temperatures on the left and right electrodes, respectively. **b**, SPVM in a 3D plot. Values projected on the bottom surface show a symmetric distribution of V_{ph} . Gray dashed lines indicate the channel-electrode interface and the solid grey line refers to the contour with zero V_{ph} . **c**, Line profiles of the V_{ph} cutting along the channel under increasing V_{ds} with grey dashed line indicating the zero V_{ph} . **d**, Voltage across the Bi_2Se_3 film versus the corresponding temperature difference to determine its room-temperature Seebeck coefficient S . The orange line is the linear fitting of the experimental data, with a coefficient of determination being 0.999. **e**, Comparison of the V_{ph} -time curve and the ΔT -time curve. **f**, Temporal response with the laser spot positioned on either side of the device, showing a positive and a negative photoresponse. **g**, Responsivity and V_{ph} as a function of incident power under 1550 nm illumination. **h**, Benchmark of the photodetector of reported PTE detectors.

On page 19 of the revised supplementary information:

Supplementary Figure 1 | SPVM determined from external biases. a,b,c, SPVM images for the V_{ds} at 50 μV (a), 100 μV (b), and 200 μV (c), respectively. d, Zero points along the channel with and without external bias. Grey dashed lines refer to the channel-electrode interface.

2. The EDS result should come from samples grown on polymer, then where does the Si element come from? By the way, the XPS should be a more precise technique for element composition analysis. The author can even do Ar⁺-etched XPS to check the composition at different thicknesses.

Reply:

We thank the reviewer for pointing out this issue. The Si element comes from the Si substrate used in the EDS characterization. This Si substrate was employed to minimize the charging effect during the measurement. The substrates are believed not to impact the film compositions significantly. This way, the compositions are expected to be consistent, ensuring the reliability of the subsequent measurement results. Moreover, to address the reviewer's concerns, we used Ar⁺-etched XPS to characterize the compositional depth profiling of the films produced by pulse irradiation synthesis (PIS) and conventional furnace annealing (CFA) (Fig. R2). The XPS characterization results are compiled in Supplementary Fig. 8f and g, consistent with the original EDS results, indicating that PIS effectively prevents the loss of Se.

Figure R2 | **a,b**, Depth profile XPS spectra of Bi 4f core levels (**a**) and Se 3d core levels (**b**) of the PIS-produced Bi_2Se_3 film. **c,d**, Depth profile XPS spectra of Bi 4f core levels (**c**) and Se 3d core levels (**d**) of the CFA-produced Bi_2Se_3 film.

Following the reviewer's suggestion, we replaced the EDS data with the more precise XPS depth profile characterization, with all changes presented below:

On line 11 of page 4 in the revised manuscript:

... At the same time, the PIS also avoids the Se sublimation loss due to its high saturated vapor pressure, which is confirmed by the compositional depth profiling of the film (Supplementary Fig. 8f and g)³¹.

On line 16 of page 7 in the revised manuscript:

... The elemental composition was obtained by the EDS module attached to the SEM-...

On line 19 of page 7 in the revised manuscript:

... For the investigation of the surface chemical state and the atomic ratio of the film, XPS was employed using a Thermo Fisher ESCLAB Xi⁺ system. The depth profiling of the elemental ratio was realized by Ar⁺-etching, with each etching depth controlled to approximately 10 nm. ...

On page 11 of the revised supplementary information:

Supplementary Figure 2 | SEM images indicate that prolonged annealing destroys the film quality. a,b, Low-magnification (a) and high-resolution (b) SEM images of the Bi₂Se₃ film after PIS. **c-e**, Larger SEM image (c) showing the degradation of the Bi₂Se₃ film after CFA and detailed morphologies under low-magnification (d) and high-resolution (e). **f,g**, XPS depth profile analyses of the Bi₂Se₃ film after PIS (f) and CFA (g).

3. The detectivity can be extremely high for 2D based detectors in visible and NIR range (>1E10 Jones). The performance shown here is just comparable with PTE type detector. The author should explain the advantage of the Bi₂Se₃ film produced by this technology compared with non-PTE type materials and also possible solutions to enhanced its performance.

Reply:

We appreciate the reviewer for this comment. The thermoelectric materials in PTE detectors typically possess higher conductivity, leading to a higher dark voltage noise and lower detectivity than some non-PTE detectors using semiconductors. In this study, we propose a unique PIS synthesis technique and apply it to synthesize thermoelectric thin film for wide-spectrum detection of PTE. The advantages of PIS-produced Bi₂Se₃ compared with non-PTE type materials come from the three following aspects:

1) PIS achieves an ultra-fast processing speed under a low temperature, unachievable

by other conventional techniques. In a typical PIS, the self-propagating combustion process triggered by an energy pulse effectively avoids high temperatures, and the resultant combustion wave rapidly passes through the thin film, allowing the material to crystallize within one second. This feature promises compatibility with various substrates. In stark contrast, non-PTE materials often require a high-temperature process (e.g., CVD, magnetron sputtering, and multi-source evaporation with post-annealing) for crystallization, thus limiting the choice of polymeric substrates, especially when mechanical conformability is desired.

2) An unreported thermal coupling mechanism is employed due to the unique features of PIS. We found that thermal diffusivities of different substrates correlate with their temperature profile under local illumination, which couples with the thermoelectric film and affects the output photovoltages. The low temperature and rapid process in PIS allow us to use flexible substrates with suitable thermal properties, which are typically thermally unstable. Therefore, we present a new pathway that further manipulates the performance of PTE devices. In contrast, the photoresponse of non-PTE materials is generally only related to the material's intrinsic properties, often neglecting the influence of the substrate.

3) The PTE photodetector based on Bi_2Se_3 realizes a wide spectral detection from visible to infrared optical communication band. The PTE effect includes two processes: the photothermal conversion and the Seebeck effect, which provides a powerful platform that transfers optical signals to electrical readouts without the limitation of the bandgap of active materials. However, the mechanism of non-PTE detectors is primarily based on the photoconductive effect or the photovoltaic effect, so the spectral response is constrained by the bandgap of semiconductors. To tackle the wavelength selectivity and achieve a wide spectral response, especially in the near-infrared range, strategies, such as narrow-bandgap semiconductors, quantum wells, and hybrid structures have been reported, but they often involve complex configurations, limiting the development of non-PTE type materials.

Apart from the enhancement of the thermal coupling by the thermal management of substrates, the performance of the PTE Bi_2Se_3 detector can be further improved by either tuning the thermal properties of the active materials or enhancing the light-matter interactions, including elemental doping, rational electrode design, plasmonic nanoantenna, hybrid structure, and so on. These strategies are expected to be compatible with PIS, suggesting their great potential. In response to the reviewer's comment, we have provided additional discussion of the advantages of Bi_2Se_3 produced by PIS and possible solutions for enhancing its performance in the revised manuscript:

On line 31 of page 2 in the revised manuscript:

... Compared to non-PTE photodetectors, our PTE-based film photodetectors produced by PIS exhibit outstanding compatibility with flexible substrates and bandgap-independent wide spectral response. ...

On line 40 of page 5 in the revised manuscript:

... The performance of the Bi₂Se₃-based PTE detector can be further improved through material doping, rational electrode design, plasmonic nanoantenna, hybrid structure, and so on⁴⁶⁻⁵⁰.

46. Saeed, Y., Singh, N. & Schwingenschlögl, U. Enhanced thermoelectric figure of merit in strained TI-doped Bi₂Se₃. *Appl. Phys. Lett.* **105**, 031915 (2014).

47. Guo, W. et al. Sensitive Terahertz Detection and Imaging Driven by the Photothermoelectric Effect in Ultrashort-Channel Black Phosphorus Devices. *Adv. Sci.* **7**, 1902699 (2020).

48. Mashhadi, S., Duong, D. L., Burghard, M. & Kern, K. Efficient Photothermoelectric Conversion in Lateral Topological Insulator Heterojunctions. *Nano Lett.* **17**, 214-219 (2016).

49. Wu, D. et al. Plasmon-enhanced photothermoelectric conversion in chemical vapor deposited graphene p-n junctions. *J. Am. Chem. Soc.* **135**, 10926-9 (2013).

50. Wei, J. et al. Mid-infrared semimetal polarization detectors with configurable polarity transition. *Nat. Photonics* **15**, 614-621 (2021).

On line 20 of page 6 in the revised manuscript:

... In a typical PIS, the self-propagating combustion process triggered by an energy pulse effectively avoids high temperatures, and the resultant combustion wave rapidly passes through the thin film, allowing the material to crystallize within one second. ...

On line 27 of page 6 in the revised manuscript:

... It is found that thermal diffusivities of different substrates correlate with their temperature profile under local illumination, which couples with the thermoelectric film and affects the output photovoltages. The low temperature and rapid process in PIS allow us to use flexible substrates with suitable thermal properties. In this regard, we present a new pathway that further manipulates the performance of PTE devices. Moreover, the outstanding wide spectrum photoresponse, high responsivity (71.9 V/W for 1550 nm), and fast response (< 50 ms) surpass most of the counterparts. The PTE effect, which includes two processes: the photothermal conversion and the Seebeck effect, provides a powerful platform that transfers optical signals to electrical readouts without the limitation of the bandgap of active materials. ...

REVIEWERS' COMMENTS

Reviewer #2 (Remarks to the Author):

All my concerns have been addressed and the overall quality of the manuscript has been improved a lot. I think the manuscript can be published.